

# Multidecadal satellite-derived Portuguese Burn Severity Atlas (1984−2022)

Dina Jahanianfard[a], Joana Parente[b], Oscar Gonzalez-Pelayo[c], Akli Benali[d]

[a] Centro de Estudos Florestais, Instituto Superior de Agronomia, Universidade de Lisboa
Forest Research Centre, School of Agriculture, University of Lisbon
[b] cE3c–Centre for Ecology, Evolution and Environmental Changes, and CHANGE Global Change and Sustainability Institute, Faculdade de Ciências da Universidade de Lisboa, Lisbon, Portugal
[c] Centre for Environmental and Marine Studies (CESAM), Department of Environment and Planning, University of Aveiro, Campus Universitario de Santiago, 3810-193 Aveiro, Portugal
[d] Centro de Estudos Florestais, Laboratório Associado TERRA, Instituto Superior de Agronomia, Universidade de Lisboa
Forest Research Centre, Associate Laboratory TERRA, School of Agriculture, University of Lisbon

Correspondence to: Dina Jahanianfard (dinaj@ua.pt)

**Abstract.** Long-term burn severity assessment can support better pre- and post-fire management plans. In this study, Portuguese Burn Severity Atlas was created containing historical fires in Portugal from 1984 to 2022. As prerequisites, fire data were gathered and delimited for all years. Due to the availability of satellite images, for different years, different imagery from Landsat sensors
(30m) were applied. Exploratory analysis showed that burn severity estimates are significantly affected by the time lag between the satellite imagery acquisition and the fire date. We explicitly incorporated the effect of time lag in the degradation of burn severity estimates in the selection of the most suitable pre- and post-fire satellite images for each fire. Using Google Earth Engine, burn severity estimates were calculated for fires equal to or larger than 500 ha between 1984 and 2000 and larger than 100 ha for fires from 2001 to 2022 with known start and end dates (*valid fires*). Different indices were calculated, such as the differenced
Normalized Burn Ratio (dNBR), relative dNBR (RdNBR), Relativized Burn Ratio (RBR), and a burn severity index that combines dNBR with enhanced vegetation index (dNBR-EVI). Overall, in Portugal, 4.92M ha burned over the 38-year period (1984−2022), from which 3.19 million ha were caused by *valid fires* (64.8 %). Among these, a total area of 3.11 million ha had burn severity estimates via the applied indices (97 % of valid and 63 % of all fires). Results show that Portugal has experienced, on average, "high" burn severity throughout this period, with large percentages of dNBR pixels between 0.419 and 0.66 (29 %) and > 0.66 (20
%). Estimates from different burn severity indices provided a more complete representation of the burn severity impacts. Via the analysis of only three fires throughout the study period, the dNBR-EVI showed potential in differentiating the "unburned" and "regrowth" burn severity while RBR was more prone to signal saturation, i.e., inability to show "high" and "very high" burn severity. However, more in-depth research is needed to fully confirm these properties. This atlas can be accessed at https://doi.org/10.5281/zenodo.12773611 (Jahanianfard et al., 2024) and be used by researchers to have a better understanding of
historical fires, their corresponding impacts on vegetation cover, air, soil, and water quality, and identification of the most influential environmental and climatical drivers of burn severity.

**Keywords**: Time lag, Google Earth Engine, Normalized Burn Ratio (NBR), Landsat, image SUITABILITY, confidence map.

## 1 Introduction

Fires are global widespread natural, dynamic, and periodically disturbing phenomena (Whitman et al., 2020; Fernández-Guisuraga
et al., 2023b; Kurbanov et al., 2022; Jain et al., 2020) with more than half of the surface land at risk of being affected (Alonso-González and Fernández-García, 2021). Fires expose terrestrial ecosystems with various impacts on forest ecology and structure,





soil erosion, loss of biodiversity, and endanger human life and infrastructures. Over the past 20 years, fires have burned on average 0.4 billion ha of land annually, with a cumulative total of 7.2 billion ha of burned area globally (Kurbanov et al., 2022).

According to statistics provided by the European Commission (2018), approximately 50,000 fires have burned from 1980 to 2018,
with an annual average of 0.5 million ha, especially in five Mediterranean European member states: Spain, Portugal, Italy, Greece, and France (Fernández-Guisuraga et al., 2023a). In these countries, the occurrence of extreme fires is getting more frequent and more intense with larger burned areas as their fire regime has shifted from "fuel-limited" to "drought-driven" (Pausas and Fernández-Muñoz, 2012). The fire regime shift has heterogeneous extent, seasonality, and frequency (Morresi et al., 2022). Its two main causes are accumulation of flammable fuels combined with consequences of global warming such as prolonged and more
frequent droughts and heatwaves (Fernández-Guisuraga et al., 2023a). The accumulation of flammable fuels are caused by land use change, agricultural farming abandonment in rural zones, and lack or absence of adaptive management (Moreira et al., 2020). Moreover, increasing global warming, will likely lead to prolonged fire seasons which may contribute to an increase in number, frequency, and area of fires (Moreira et al., 2020; Miller et al., 2023; Holsinger et al., 2021; Fernández-Guisuraga et al., 2021). However, it is still unclear whether this predicted increase will lead to an increase in burn severity (Soverel et al., 2011; Morresi
et al., 2022; Fernández-Guisuraga et al., 2023b; Parks et al., 2016).

Burn severity can be defined as the extent to which fire induces ecological and visible changes on soil and vegetation (Key and Benson, 2006; Key, 2006; Lentile et al., 2006; Veraverbeke et al., 2010). Estimation of burn severity provides insights into forming better pre- and post-fire management strategies, including fuel treatments and post-fire recovery plans (Chu and Guo, 2014; Miller et al., 2023; García-Llamas et al., 2019). Burn severity estimates are highly time-sensitive since post-fire conditions depend on
pre-fire conditions (Miller et al., 2023). A delayed estimation of burn severity will most likely lead to its poor estimation due to first environmental responses such as forest recovery, tree/seedling recruitment, resprouting, vegetation regrowth, and ashes wash off by wind or precipitation (Miller et al., 2023; Dos Santos et al., 2020; Keeley et al., 2008; Keeley, 2009; Chu and Guo, 2014; Key, 2006) or secondly by seasonal lag, influenced by low sun angle, increasing the risk of shadow contamination even on flat terrains (Holsinger et al., 2021; Key, 2006). Hence, burn severity can be assessed during three periods based on different time lags,
which are the difference between the dates of fire occurrence and burn severity estimation, categorized as rapid assessment (less than two weeks), initial assessment (1 to 8 weeks) and extended assessment ( 2 to 12 months) (Key, 2006).

The most reliable approach to estimate burn severity is via field assessment (Key and Benson, 2006) by measuring the observable fire-induced changes such as the extent of fire-consumed vegetation, stems of vegetation being charred, soil being exposed, and loss of chlorophyll in leaves (Keeley, 2009). These fire-induced changes correspond to structural, thermal, and spectral alterations
in soil and vegetation (Miller et al., 2023). One of the most used metrics is the Composite Burn Index (CBI) (Key and Benson, 2006; García-Llamas et al., 2019) that visually ranks the burn severity from 0 (unburned) to 3 (high severity) (Parks et al., 2018; Fernández-Guisuraga et al., 2023a; Addison and Oommen, 2018). The reason for CBI's popularity is due to its rapid protocol and its overall estimation of fire induced damage on vegetation and soil, especially when assessing burn severity of large fires. However, burn severity field assessments have multiple drawbacks as they are intensive, logistically challenging, highly resource-
dependent, especially in inaccessible and/or remote burned areas, and have limited capability in capturing the burn severity heterogeneity over large burned areas. Moreover, the impacts of historical fires and evolution of burn severity cannot be measured via field assessment (Miller et al., 2023).

The emergence of remote sensing (RS), especially via application of satellite imagery, over the past decades has enabled free-of-charge remotely-sensed burn severity assessment as an alternative option to expensive and time-consuming field severity (Miller
et al., 2023; Holsinger et al., 2021; Fernández-Guisuraga et al., 2021; Miller and Thode, 2006; Parks et al., 2014; Fernández-García et al., 2018). The capability of capturing spectral information in the visible, near-infrared (NIR) and shortwave infrared (SWIR)



parts of the electromagnetic spectrum (Key and Benson, 2006) has enabled the detection of fire-induced structural, thermal, and spectral changes on land surface (Miller et al., 2023). Satellite sensors have different optical bands, spatial resolutions, temporal revisiting frequencies and time spans (Lentile et al., 2006). Thus, there are tradeoffs in the application of different satellite RS

sensors and the availability of clear-sky imagery (Miller et al., 2023). Moreover, caution should be taken when using RS products as they may acquire top of the canopy reflectance with limited capability to estimate burn severity of the understory strata (García-Llamas et al., 2019; Mihajlovski et al., 2023). Last but not least, RS-derived burn severity estimates "must be linked to ground-truth data" (García-Llamas et al., 2019; Miller et al., 2023; Chu and Guo, 2014).

Multiple studies have found moderate correlations between satellite-derived RS and ground burn severity indices, providing higher
confidence in RS derived burn severity estimates. However, the strength of these correlations varies from one region to another and can be influenced by environmental factors such as fuel, vegetation type, and topography (Fernández-Guisuraga et al., 2021). In this context, mono (only post-fire image) and bi-temporal (both pre-and post-fire images) normalized burn ratio (NBR) derived indices, such as differenced Normalized Burn Ratio (dNBR), Relative Differenced Normalized Burn Ratio (RdNBR), Relativized Burn Ratio (RBR), and burn severity index that combines dNBR with enhanced vegetation index (dNBR-EVI) (Gao et al., 2000),
have been applied. The dNBR (Key and Benson, 2006) is considered as the "standard" index for burn severity quantification (Alonso-González and Fernández-García, 2021), specifically in the Mediterranean regions (Fernández-García et al., 2022; Miller and Thode, 2006; Picotte et al., 2016; Fernández-Guisuraga et al., 2023a; Chu and Guo, 2014; Keeley et al., 2008; Fernández-García et al., 2018). The RdNBR provides a relative measurement of burn severity based on the pre-fire state of vegetation (Miller and Thode, 2006) and has proven to be more sensitive than the dNBR, especially in areas with low vegetation cover density (Parks
et al., 2014). However, the calculation of RdNBR presents some difficulties due to its formula and its "numerically unstable" range as the result of pre-fire NBRs with very low values (Fernández-Guisuraga et al., 2023a). Another relative measure of burn severity without the calculation difficulties is RBR (Parks et al., 2014). According to Fernández-Guisuraga et al. (2023a), RBR showed better correlation with CBI in Mediterranean ecosystems in comparison to dNBR. Additionally, according to Fernández-García et al., (2018), their proposed index known as dNBR-EVI, exhibits the best correlation with CBI in Mediterranean regions in
comparison to NBR, dNBR, RdNBR, and RBR. Moreover, dNBR-EVI is claimed to show no signal saturation in high-severity areas, as saturation is a known issue for NBR-derived indices in regions with high burn severity (Fernández-García et al., 2018; Santis et al., 2010; Fernández-Guisuraga et al., 2023a).

The estimation of burn severity of historical fires can be performed using satellite-derived RS indices. This feature can enable the evaluation of changes or trends in burn severity patterns over a specific period (Lutz et al., 2011; Picotte et al., 2016). The first
project devoted to the creation of a burn severity atlas was the Monitoring Trends in Burn Severity (MTBS) which provided dNBR and RdNBR maps for large fires from 1984 to present in the USA using Landsat imagery (Eidenshink et al., 2007; Picotte et al., 2020). There have been records of burn severity atlases created for parts of some countries such as Canada (Picotte et al., 2016; Whitman et al., 2020; Guindon et al., 2021). Moreover, the MOSEV dataset provides 20 years of burn severity maps using Moderate Resolution Imaging Spectroradiometer (MODIS) imagery and products (Alonso-González and Fernández-García, 2021).
Although this dataset provides daily global coverage, it has considerable limitations such as the spatial resolution (500m), the limited capability of mapping burn severity heterogeneity especially at regional scale (Alonso-González and Fernández-García, 2021), the absence of burn severity estimates for fires before the year 2000 and low burned area mapping accuracy (Moreno-Ruiz et al., 2020).

To the best of our knowledge, detailed long-term estimates of burn severity are missing for European fire-prone countries, such as
Portugal. Thus, the main objective of this study is to create a high resolution multidecadal burn severity atlas for mainland Portugal entitled "Portuguese Burn Severity Atlas".



## 2 Data and methods

The study area consists of mainland Portugal ("37°N to 42°N latitude and 6°W to 10°W longitude") (Parente et al., 2016), covering around "90,000 km² of Southern Europe" (Rego and Bacao, 2010) generally with Mediterranean climate consisting of "warm, dry

summers and cold, wet winters" (Nunes et al., 2016), elevation ranging "from sea level to approximately 2000m"(Mora and Vieira, 2020), and with domination of different vegetation types within its extent (e.g., "farmland and evergreen oak (*Quercus suber, Q. rotundifolia*) woodlands in South, forests of pine (*Pinus pinaster*) and eucalypt (*Eucalyptus globulus*) in North and shrublands and deciduous oak forests" in Center) (Tonini et al., 2017).

To estimate the burn severity of historical fires in Portugal (1984−2022), it is necessary to primarily gather *fire data* containing the

start and end dates (Sdate and Edate), the burned perimeters and extents (Sect. 2.1). Then, we selected RS sensor or family of sensors to have coherency over the study period (Sect. 2.2). Next steps were to select burn severity indices well correlated with ground observations specifically in the Mediterranean regions (Fernández-Guisuraga et al., 2023a)  (Sect. 2.3), assign sampling period to select RS imagery (Sect. 2.4), quantify the influence of time lag on burn severity estimates and accordingly apply the necessary adaptation to the sampling period (Sect. 2.5)  and finally to calculate the burn severity estimate for each of the fires with

the most suitable pair of images (Sect. 2.6).

### 2.1 Fire data

We focused the work on fires larger than 100 ha, that were responsible for 75 % of the total burned area in Portugal (Divisão de Defesa da Floresta Contra Incêndios (DGRF), 2006; Fernandes, 2009). Only fires with known start and end date were kept and considered as *valid*.

For the period from 1984 to 2000, uncertainties regarding the fire dates are much larger than the subsequent years up to present. For this period, there are two official fire datasets in Portugal gathered and provided by the Instituto da Conservação da Natureza e das Florestas (ICNF), (2021): i) Portugal Rural Fire Dataset (PRFD) and ii) the National Mapping Burnt Area (NMBA) (Kanevski and Pereira, 2017). Parente et al. (2016) presents comprehensive details on both of these datasets, however in summary, the PRFD dataset contains fire dates obtained through ground measurements with exclusion of fire perimeters, while NMBA is based on RS

imagery (Parente and Pereira, 2016; Parente et al., 2018). To minimize these uncertainties, we initially intersected NMBA and PRFD. For the intersecting areas, fire dates were retrieved from the PRFD dataset. In cases of no intersections, a 10 km buffer was created around NMBA perimeters, and the buffered dataset was again intersected with PRFD. For any intersected areas, respective fire dates from PRFD were retrieved and recorded. These fire dates were manually confirmed analyzing Landsat 1 to 5 imagery acquired approximately on or as close as possible to the fire dates recorded in PRFD. In cases where the fire scars were not visible

on the "false-color composite" (R: SWIR, G: NIR, B: red) or "false- color NIR composite" (R:NIR, G: red, B:green) of Landsat 1 to 5 imagery, images taken shortly before and after the recorded dates were examined, and the corresponding fire date was updated with the acquisition date of the image where the fire scar first appeared. Additionally, duplicates and entries without geometric perimeters were removed from the fire data. Thus, although there are data of fire perimeters since 1975 (Oliveira et al., 2011), due to lack of fire dates, we determined the starting year of this atlas to be 1984. Thereafter, it was observed that mainly (98.6 %) valid

fires with areas equal to or larger than 500 ha with recorded dates were obtained for fires from 1984 to 2000.

To gather fire data from 2001 to 2022, multiple sources were utilized. The fire perimeters were supplied by the Instituto da Conservação da Natureza e das Florestas (ICNF), (2021b). These perimeters were derived through semi-automatic supervised classification of satellite imagery, with subsequent manual editing for refinement (Oliveira et al., 2011). Any discrepancies were identified and rectified by comparing the mapped fire perimeters with field statistics at the national level. The Sdates and Edates





for most of the fires were also obtained from the ICNF. However, after conducting an exploratory analysis, errors were detected in the provided fire dates. Therefore, the Sdates and Edates were redetermined by combining data MODIS and the Visible Infrared Imaging Radiometer Suite (VIIRS) thermal anomalies following the method developed by Benali et al. (2016). The data were cross-referenced with data reported by the ICNF. Moreover, visual analysis of images from various sensors, particularly for fires occurring after 2017, were also incorporated to confirm and/or update the fire dates. The minimum mapping unit for this period

(2001 to 2022) is 100 ha.

Between 1984 and 2022, a total number of 38,700 perimeters representing historical fires with a total burned area of 4.92 million ha were recorded in mainland Portugal – 2.00 million ha between 1984 to 2000 and 2.92 million ha between 2001 to 2022. Within this atlas, 3200 perimeters (8.4 %) are considered valid fires, accounting for a total burned area of 3.19 million ha (64.8 %) – 0.78 million ha between 1984 to 2000 and 2.41 million ha between 2001 to 2022 – in all vegetation types, distributed within the extent

of the mainland Portugal.

## 2.2 RS imagery: access and processing

Portuguese Burn Severity Atlas spans across several years (1984−2022) overlapping the acquisition period of several sensors. Atmospherically corrected surface reflectance images from the Landsat series of sensors were used as reference to calculate burn severity indices. This choice was based on Landsat's long available data archive, especially Landsat-5 going back as far as 1984,

the high spatial resolution (30 meters), consistency between sensors with revisiting frequency of 16 days, and provision of required bands for burn severity indices (NIR, SWIR, red, and blue bands – Sect. 2.3).

For fires from 1984 to 2001 and 2003 to 2011, imageries from Landsat-5 Thematic Mapper (TM) were applied. For 2002, both images from Landsat-5 TM and Landsat-7 Enhanced Thematic Mapper Plus (ETM⁺) were available, however, it was observed that there were more available clear-sky images from Landsat-7 in comparison to Landsat-5 imagery. Thus, for 2002, imagery from

Landsat-7 was used to estimate burn severity. For 2012, there is no Landsat imagery available except for Landsat-7 that suffered a technical failure in its scan line corrector (SLC) in May 2003 resulting in multiple gaps within its imageries since then (Key and Benson, 2006) that reduces the quality and availability of satellite imagery. Thus, to have coherency of having burn severity estimates over all the years, for only 2012, we used atmospherically corrected surface reflectance imagery of Terra abroad MODIS, with spatial resolution of 500m. For years 2013 to 2022, imageries from Landsat-8 Operational Land Imager (OLI) were applied.

In Table 1, the applied sensors for each year are summarized.

In the past, conduction of bi-temporal burn severity estimates, such as dNBR, by manually gathering pre- and post-fire images was difficult and time-consuming. However, with the emergence of cloud-based processors such as Google Earth Engine (GEE), through "an internet-based application programming interface (API) written in JavaScript" (Perez and Vitale, 2023), this process is now feasible, free and can be semi-automated (Yilmaz et al., 2023; Whitman et al., 2020) when fire data are available (Parks et

al., 2018). In this study, all the processes of image acquisition, calculation of the burn severity indices, and generation of burn severity maps were performed within the GEE platform. One of the biggest limitations of GEE is its optimization when performing heavy processing (Carille et al., 2024). To overcome this limitation, separate functions were defined in our code to process images and generate burn severity estimates. The GEE datasets of different Landsat sensors are also summarized in Table 1.

**Table 1.** Summary of sensors used for each year and their characteristics (availability date, corresponding bands, GEE dataset). **

With gaps within its imagery since May 2003.



| Years | Sensor | Availability date | GEE dataset | Bands |
|---|---|---|---|---|
| 1984−2001 and 2003−2011 | Landsat- 5 (TM) | 16 March 1984 to 05 May 2012 | LANDSAT/LT05/C02/T1_L2 (USGS Landsat 5 Level 2, Collection 2, Tier 1\|Earth Engine Data Catalog\|Google Developers, 2021) | NIR:'SR_B4' SWIR: 'SR_B7' RED: 'SR_B3' BLUE: 'SR_B1' Cloud-mask: 'QA_PIXEL' |
| 2002 | Landsat- 7 (ETM[+]) | 28 May 1999 to 26 September 2023 ** | LANDSAT/LE07/C02/T1_L2 (USGS Landsat 7 Level 2, Collection 2, Tier 1 \| Earth Engine Data Catalog \| Google Developers, 2023) | NIR:'SR_B4' SWIR: 'SR_B7' RED: 'SR_B3' BLUE: 'SR_B1' Cloud-mask: 'QA_PIXEL' |
| 2012 | MODIS - Terra | 18 February 2000 to present | MODIS/006/MOD09A1 (MOD09A1.006 Terra Surface Reflectance 8-Day Global 500m \| Earth Engine Data Catalog \| Google Developers, 2023) | NIR: 'sur_refl_b02' SWIR: 'sur_refl_b07' RED: 'sur_refl_b01' BLUE: 'sur_refl_b03' Cloud-mask: 'StateQA' |
| 2013–2022 | Landsat- 8 (OLI) | 18 March 2013 to present | LANDSAT/LC08/C02/T1_L2 (USGS Landsat 8 Level 2, Collection 2, Tier 1 \| Earth Engine Data Catalog \| Google Developers, 2023) | NIR: 'SR_B5', SWIR: 'SR_B7' RED: 'SR_B4' BLUE: ,'SR_B2' Cloud-mask: 'QA_PIXEL' |

**2.3 RS burn severity indices**

Portuguese Burn Severity Atlas is created using bi-temporal NBR derived burn severity indices as dNBR, RdNBR, RBR, and
dNBR-EVI. The NBR is calculated via the normalized difference of NIR and SWIR optical bands based on the principle that
healthy vegetations have high NIR reflection while burned/dead vegetations have high SWIR reflection (Key and Benson, 2006;
Whitman et al., 2020). The burn severity indices, along with their formulas, are summarized in Table 2.

The atlas includes the corresponding offset values, i.e. mean value for each of these indices outside the burned area, that are
representative of the unburned environment surrounding the burned areas. They are incorporated to isolate fire induced changes
from unburned environment (Key, 2006; Miller and Thode, 2006; Parks et al., 2014, 2018), minimize the impacts of differences in
pre- and post-fire imagery due to phenology or precipitation conditions (Parks et al., 2018), and improving the comparison of burn
severity estimates across fires (Parks et al., 2014). The offset was estimated by calculating the mean values of pixels located within
180 m outside the burned area for all the years following the formula of each corresponding burn severity index (Parks et al.
(2018)). However, a buffer of 500 m was used for 2012 because only MODIS data was available for this year (see Sect. 2.2).

**Table 2.** Summary of burn severity indices and their corresponding formulas. NIR, SWIR, RED, and BLUE refer to the satellite
bands of NIR, SWIR, red, and blue bands, respectively. *The offset refers to the corresponding mean value of the burn severity
index within the buffer (180 m except for year 2012 which is 500 m) outside the burned area.



| Spectral burn severity index | Formula | Interpretation | Reference |
|---|---|---|---|
| **NBR** | $\dfrac{NIR - SWIR}{NIR + SWIR}$ | Demonstrating the vegetation loss based on the principle that healthy vegetations have high NIR and burned ones have high SWIR reflections. | (Key and Benson, 2006; Whitman et al., 2020) |
| **dNBR** | $NBR_{pre} - NBR_{post} - offset^*_{dNBR}$ | Absolute difference of pre- and post-fire state of vegetation. | (Key and Benson, 2006) |
| **RdNBR** | 1. $\dfrac{dNBR}{\sqrt{|NBRpre|}} - offset^*_{RdNBR}, |NBR_{pre}| \geq 0.001$ <br> 2. $\dfrac{dNBR}{\sqrt{|0.001|}} - offset^*_{RdNBR}, |NBR_{pre}| < 0.001$ | Relative difference of pre- and post-fire state of vegetation, considering the pre-fire state of vegetations and their density. | (Miller and Thode, 2006; Parks et al., 2018) |
| **RBR** | $\dfrac{dNBR}{NBRpre+1.001} - offset^*_{RBR}$ | Relative difference of pre- and post-fire vegetation state without the difficulty in its formula. | (Parks et al., 2014) |
| **dNBR-EVI** | $EVI = 2.5 \times (\dfrac{NIR+RED}{NIR+6*RED-7,5*BLUE+1})$ <br><br> $dNBR + EVI_{post} - offset^*_{dNBR-EVI}$ | Demonstrating the amount of vegetation loss without the saturation of pixels in areas with high burn severity. | (Gao et al., 2000) |

### 2.4 RS imagery sampling period

Considering the fact that the Landsat series have a revisit frequency of 16 days, few clear images annually are available in cloudy regions, especially with older satellites like Landsat-5 (Gao et al., 2006). Thus, a longer sampling period is often needed, while

there is no guaranteed window. The probability of clear images during rapid assessment (<two weeks) is low. In the initial assessment (one to eight weeks), clear Landsat images may be obtained, but the probability is not high, and assessing delayed consequences like tree mortality or tree survivorship is challenging (Key, 2006). In the extended assessment period (two to twelve months), the probability of acquiring clear images is higher, however, as mentioned above, RS-derived burn severity indices are strongly affected by time lag (Morresi et al., 2022).

To address these issues, we set our test sampling periods as follows: one day to 120 days before each fire's Sdate and three days to 120 days after each fire's Edate as the pre- and post-fire sampling periods, respectively. The one-day before Sdate and three-days after Edate were defined to avoid images with active fires or smoke contamination. To minimize seasonal differences, we capped the sampling period at 120 days, though some variation in seasons may still have occurred.

### 2.5 Quantification of time lag influence

To produce the most accurate and representative burn severity estimates from satellite imagery for each fire, the most suitable pre- and post-fire images are required, meaning images with the lowest time lag or with lowest difference between fire event and their acquisition date. Hence, it is crucial to understand and quantify the impact of the time lag on burn severity estimates.

Valid fires from 2013 to 2018 were chosen to be analyzed and the Landsat-8 (OLI) imagery was used. Primarily, for each fire, two image collections (IC) were created in GEE. IC is a GEE data type that stores a set of images taken within the bounds of any area

of interest within the assigned sampling period (Carille et al., 2024). In this case, corresponds to images overlapping the burned area during our sampling period (each temporal buffered fire date ± 120 days) resulting in one IC for pre- and one IC for post-fire images (preIC and postIC, respectively). Each IC was filtered to only have images with actual coverage over at least 90 % of the burned area and at least 90 % of the covering pixels were cloud/cirrus free not only over the burned area but also within a 2-km buffer around it. This filter was used to account for contaminated pixels within the border and to exclude the impacts of shadow





contamination of cloud/cirrus within the burned area surroundings. The NBR of all the available processed pre- and post-fire images were calculated, hereafter referred to as pre-NBR and post-NBR, respectively.

For each fire, the dNBRs from all the possible combinations of pre- and post-NBRs were calculated in MATLAB R2021b. A subset of fires was created with at least one dNBR estimation with both pre- and post-NBRs with time lags equal to or less than 7 days with lowest cloud contamination, referred to as the *reference dNBR*. We assigned this threshold under the assumption that

significant dNBR degradation was unlikely to occur within 7 days. To quantify the variation in dNBR caused by time lag, the difference between the reference dNBR and the lagged dNBRs, calculated by either lagged pre- or/and post-NBRs, were estimated at a pixel scale- with corresponding results shown in Sect.3.2 and Figure 4. The time lag of both pre- and post-NBRs used for reference dNBR calculation were considered as the basis of dates instead of fires' Sdates and Edates.

Using simple linear regression analyses, we modelled the median of pixel-by-pixel dNBR difference (dependent variable) as

function of pre- or post-fire time lags (independent variables). After 110 days, the dNBR degradation was too high (higher than 0.1), leading to inaccurate burn severity estimates (different discrete dNBR classification according to European Forest Fire Information Service (hereafter as EFFIS) (European Commission, 2018; Llorens et al., 2021). Thus, we reduced our sampling period to ±110 days. From the found correlation and adaptation of our sampling period, a function represented in Eq. (1) was developed to calculate "SUITABILITY" property that penalizes possible RS images based on their time lag (ranging from 0 to 100

for time lags of 111 and 0 days, respectively).

$$\text{Suitability [\%]} = 1 - (\text{time lag} \times 0.009) \tag{1}$$

**2.6 Burn severity calculation**

RS burn severity indices were generated using pre- and post-NBRs derived from images with the highest "SUITABILITY".

Thereafter, it was observed that a significant number of fires had a large proportion of their areas with missing values. To address this issue, an "iteration process" was introduced. Within this process, following the computation of the burn severity indices with the pair of images with highest SUITABILITY, a comparison was made between the area with dNBR estimates and the original burned area. For fires with a missing data extent larger than 70 ha, the burn severity indices were recalculated with the pairs of images with the lower SUITABILITY, filling the missing data areas. We performed multiple trials and the lowest possible value

obtained was 70 ha without GEE code freezing during an acceptable amount of computation time. To optimize GEE performance, a maximum of four iterations were set, thus areas without burn severity estimates after the fourth iteration were disregarded (Fig.1). Details of the scenarios used for the iteration process are shown in Fig A1 along with examples of each of these scenarios presented in Fig A2 to FigA5 in the Appendix. Additionally, a "confidence" map was generated for each fire defined as the average SUITABILITY of the pair of images used to estimate the burn severity metric in a given pixel. That annual burn severity maps are

provided in absolute scale, along with their associated confidence maps.




**Figure 1.** Flowchart demonstrating the iteration process used for the calculation of burn severity indices and to have estimates for the largest proportion of the fire area as possible. PreIC and postIC are our abbreviated versions of pre-fire ImageCollection and post-fire ImageCollection, respectively. Sorted-pre and -postICs were sorted based on SUITABILITY of images from highest to lowest.

# 3 Results

## 3.1 Overview

Coverage of Portuguese Burn Severity Atlas corresponds to 3.11 million ha accounting for 63 % and 97 % of all and valid fires, respectively. From 1984 to 2000, 0.75 million ha of burned areas have burn severity estimates (15 % of all 23 % of valid) and while for fires of 2001 to 2022, 2.35 million ha of burned area have burn severity estimates (48 % of all and 74 % of valid). The



use of several iterations improved the coverage of the atlas by 14 %, adding 0.44 million ha of estimated burn severity extent to the 2.67 million ha obtained by using only one iteration (details are summarized in Table B1, presented in Appendix).

Figure 2(a) represents the spatial distribution of annual burn severity estimates of the valid fires between 1984 to 2022, using dNBR as the standard burn severity index. For pixels that burned several times, the average dNBR value is presented. To facilitate the interpretation of the burn severity estimates, color classifications were applied on dNBR pixel values according to the thresholds assigned by EFFIS (European Commission, 2018; Llorens et al., 2021). Different burn severity classes can be observed, highlighting its heterogeneity throughout Portugal. High dNBR values (>= 0.42) are concentrated in Southern and southwestern

parts of the Centro region, northeastern Vale do Tejo, and the Algarve region. Very low dNBR values (between -0.1 to 0.1) are mainly distributed in northeastern Centro and scattered in Norte regions, with isolated observations in Alentejo and Vale do Tejo. The histogram of dNBR pixel counts of all the years (without pixel averaging) along with the corresponding Cumulative Distribution Function (CDF) - secondary y-axis - and mean dNBR value are presented in Fig 2(b). In Portuguese Burn Severity Atlas, "Unburned or Regrowth of vegetation" class with dNBR <= 0.1, represents 15 % of the total dNBR pixel values. Most of

the area had burn severity estimates with dNBR values ranging from 0.2 to 0.7, with the highest associated with "High" burn severity (29 %). The mean dNBR pixel value is 0.417 with corresponding CDF of 0.61 that can be interpreted as "High" burn severity and 20 % had "Very high" burn severity with CDF of 0.9 at dNBR pixel value equal to 0.7. Thus, results show that Portugal has had, on average, "High" burn severity throughout this 38-year study period.

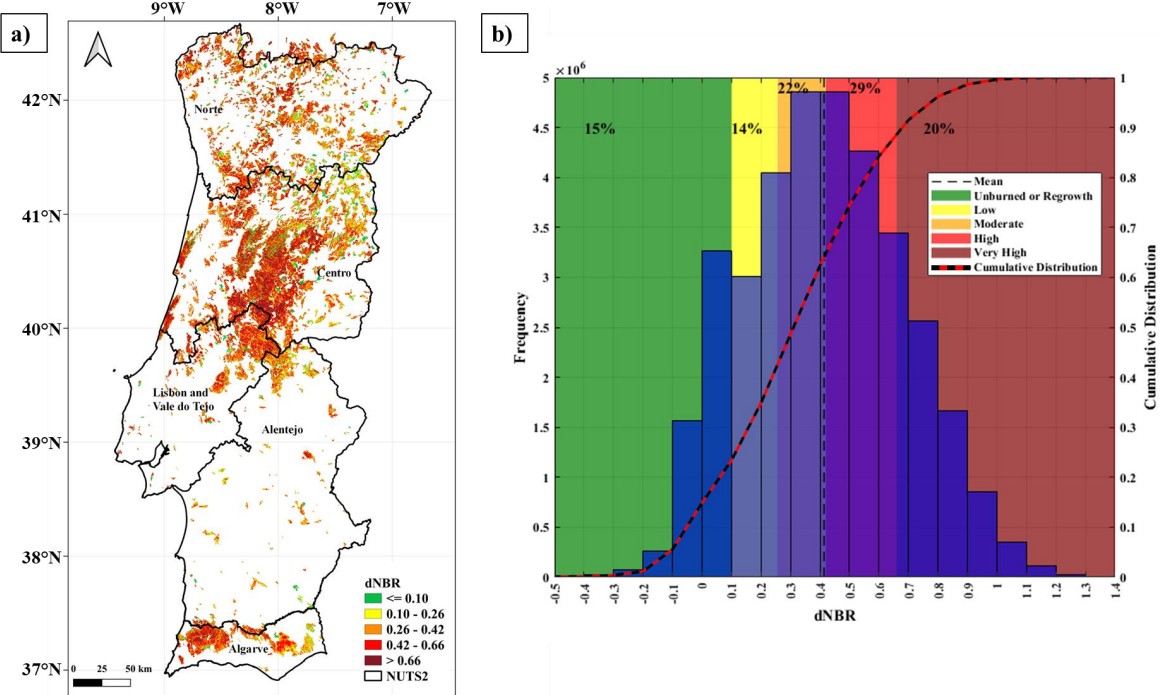

**Figure 2.** (a) Spatial distribution of the overlaid dNBR pixel value of all the years (1984 to 2022) is presented, with the average pixel value presented in the area where burned more than once. "Nomenclature of Territorial Units for Statistics" (presented as NUTS2) frontiers (Registo Nacional de Dados Geográficos - Direção-Geral do Território (DGT), 2024; Meneses et al., 2018) was applied to demonstrate the extent of the mainland Portugal and its five regions. (b) The histogram shows the distribution and frequency of all annual dNBR pixel counts, with no averaging. On the secondary y-axis, the cumulative distribution of the dNBR

pixel value is presented. The dNBR classification according to EFFIS (European Commission, 2018; Llorens et al., 2021) combined with percentage of pixel counts within these classes are also shown (panel (b)).

To have a temporal overview, in Fig 3, the annual burned extent is shown in red bars. Although there is no apparent trend, the largest burned extents were registered in 2003, 2005 and 2017, representing the largest fire seasons with a large difference in comparison to 1980's and 1990's. The highest percentage of valid burned area occurred in 2017, for which 99.8 % had burn severity

estimates. For most of the years, this atlas provides burn severity estimates for more than 90 % of area with the exception of 2007, 2006, 2011, and 1987 with values varying between 68.7 % and 86.2 %. The variation of area with burn severity estimates (pink line) highlights the lack of burn severity estimates for annual valid fires. For recent years – 2013 onwards – this variation is almost constant (on average ≈98.9 %).

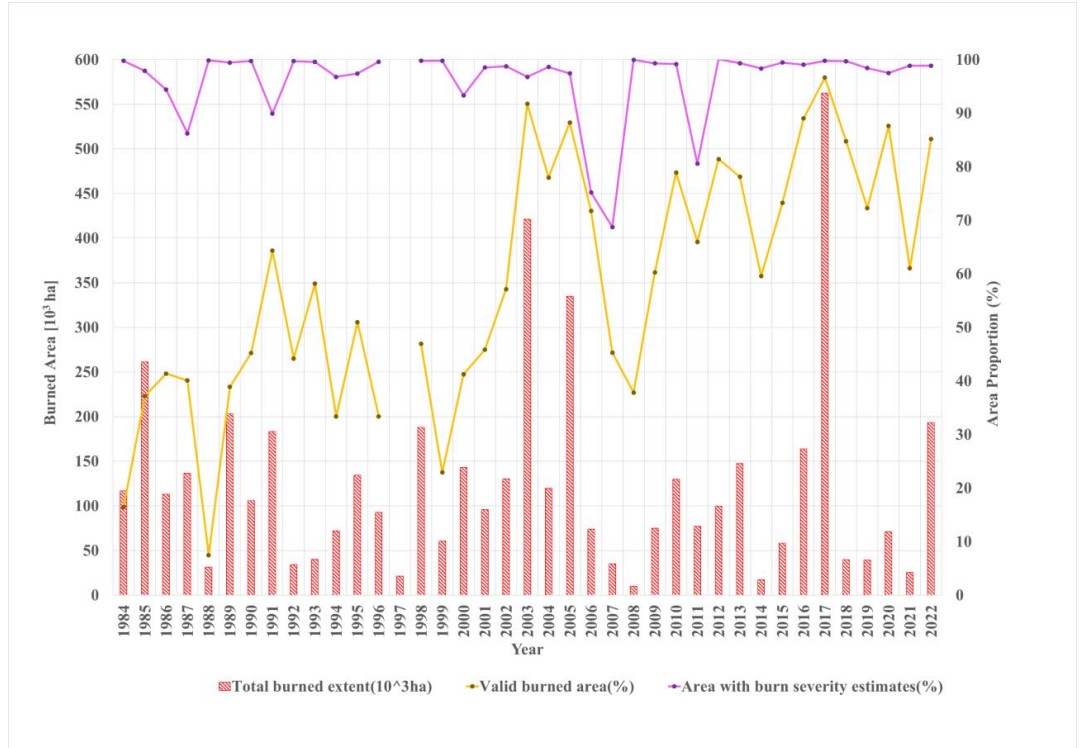

**Figure 3.** Timeline of total burned extents of fires 1984 to 2022. On left-y axis, annual burned area in unit of $10^3$ ha and on secondary y-axis the percentages of valid burned area and area with burn severity estimates are shown. No records for fires in 1997 were obtained and thus this year is presented in blank.

### 3.2 Influence of time lag on dNBR estimates

Figure 4 shows the pixel-by-pixel variability of the difference between lagged dNBR and reference dNBRs. With the increase of

both pre- and post-fire time lags, their variability increases and hence, the degradation of the dNBR estimates increases. The magnitudes of the impacts of pre- and post-fire lags are different as with the increase of post-fire time lag, the variability of dNBR difference is larger than with the increase of pre-fire time lag. On average, the increase in pre-fire time lags leads to positive differences, which means dNBR tends to be overestimated, and with increasing post-fire time lags, dNBR tends to be underestimated.





The linear regressions using time lag and the dNBR difference had $R^2_{pre}$= 0.76 and $R^2_{post}$ = 0.53, with similar slope =0.0009 and p-value <0.01. A slope of 0.0009 means that, on average, dNBR degrades by 0.0009 for each added lag day. As the slopes were similar for both pre-and post-fire, a single suitability function was adopted. Both regressions had very small (near zero) offset values (-0.012 and 0.001, respectively) and hence, they were not used in the suitability function.

The fire-by-fire confidence variability of burn severity estimates through each iteration is shown in Fig 5. No trend regarding the

variation of confidence throughout the atlas years is observed. On average, high confidence values (>80 %) for the first iteration were obtained for most of the years except for 1986 and 2010, for which Landsat-5 TM images were used (see Table 1). The range of confidence variability of the first iteration in 2007 is the largest. As expected, for the second to fourth iterations, the variability ranges are higher than the first iteration and on average the lowest confidence values (< 70 %) were observed for 2001, 2000, 2011, and 1986. Two isolated fires showed confidence values <30 % - for the second to fourth iterations- in 2010 and 1985 and one with

confidence <20 % for the first iteration in 1986. Concurrently, the fire-by- fire variability analysis performed on both pre- and post-fire time lags of the first and second to fourth iterations are following these results. Additionally, this analysis highlighted that on average both pre- and post-fire time lags from the first iterations were less than ±50 days. For recent years – 2013 onwards and especially via the first iteration - both pre- and post-fire time lags on average are less than ±20 days (fire-by- fire boxplot is presented in Fig B1, presented in Appendix).

Due to the coarse spatial resolution of MODIS Terra (500 m) and the 70 ha enforced threshold for the iteration process, in 2012, the burn severity estimates were obtained only through the first iteration as no more iterations were needed. Moreover, due to its high temporal resolution, with daily revisiting frequency, the suitability of images used for burn severity estimates were high. Thus, for 2012, the variation of confidence is small and its average is high (>90 %).

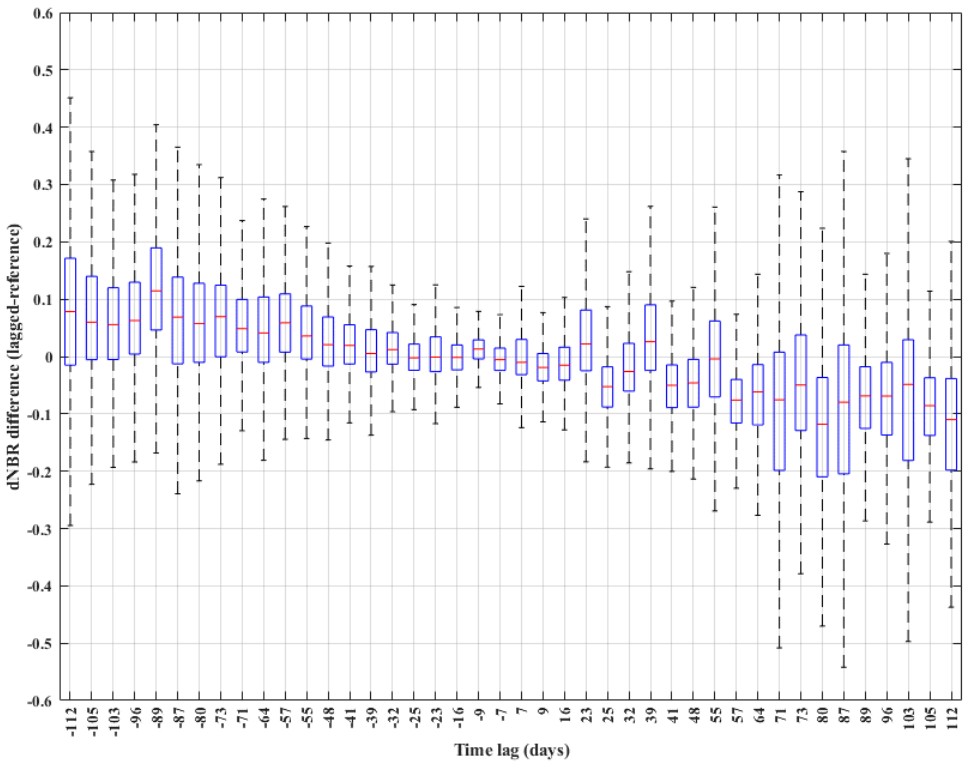



**Figure 4.** Boxplot presents dispersion of pixel-by-pixel difference of dNBRs versus the time lag. The time lag, calculated as the difference of time lags of reference NBRs, is shown on the x-axis and the negative values represent pre-fire time lags, while the positive ones represent the post-fire time lags.

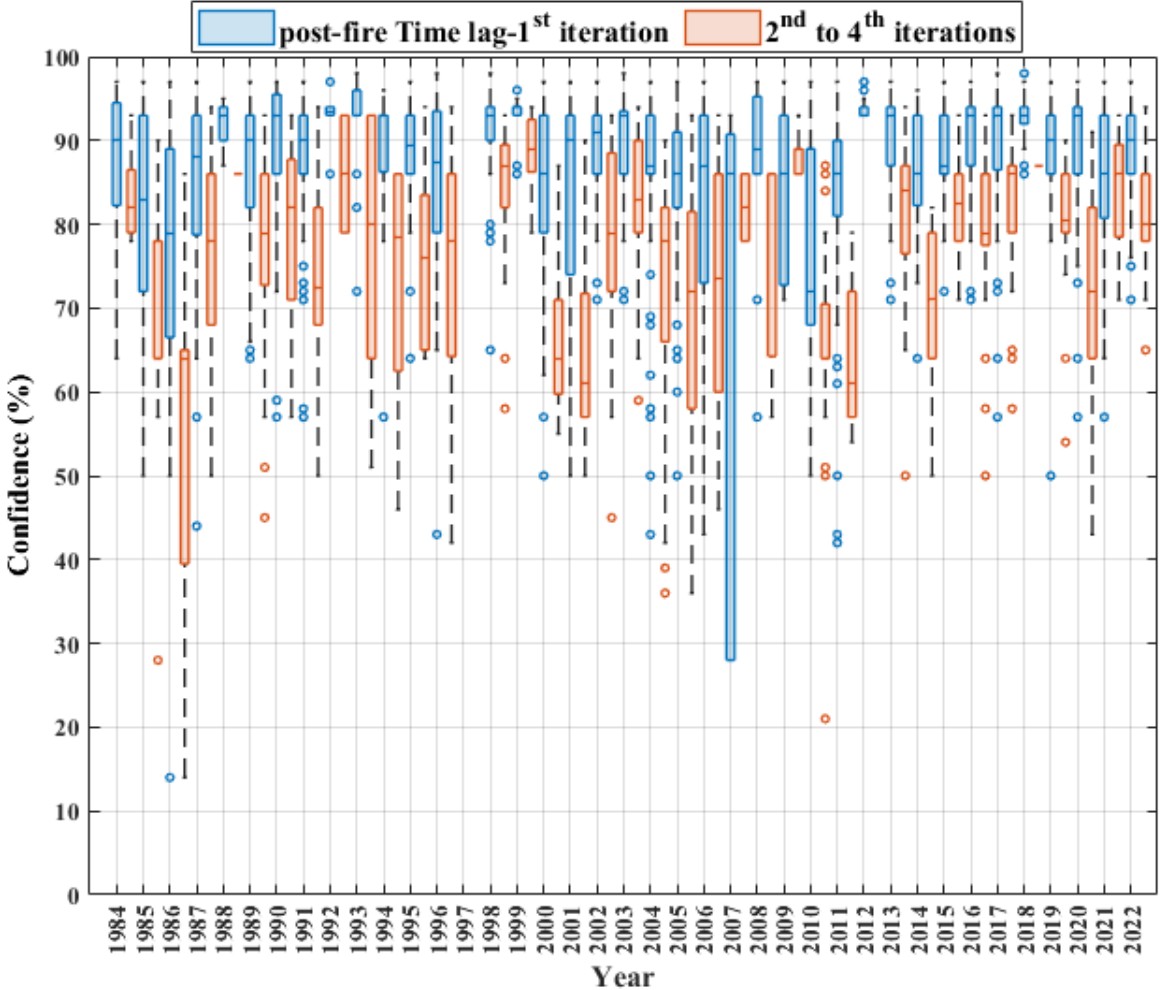

       **Figure 5.** Fire- by-fire confidence variability over the years obtained through the first iteration or second to fourth iterations. In
2012, burn severity estimates were only obtained through the first iteration. The year 1997 is presented in blank due to not having valid fire data.



### 3.3 Burn severity according to the different indices

The burn severity estimates of three very large fires (area > = 10,000 ha) that occurred in 2017, 2003, and 1991 with different
indices (dNBR, RBR, RdNBR and dNBR-EVI) are provided in Fig 6- (a.1), (b.1), and (c.1), as examples. Overall, dNBR and
dNBR-EVI have very similar distributions, while RdNBR has the highest range in contrast to RBR, suggesting having higher
sensitivity to burn severity variation.

Throughout these figures, patches with low dNBR pixel values (<0.1) were coincident with low values for the different burn
severity indices except for the dNBR-EVI, where these patches had pixel values mostly ranging from 0.1 to 0.256. However, there
is a large green patch in Fig 6(c.1) in the eastern part of the fire perimeter even with dNBR-EVI, which suggests an error in the
burned area mapping. Thus, only based on these examples, results suggest that with dNBR-EVI index and its low pixel values
(<0.1), the "unburned" class can be distinguished, only under the assumption that dNBR-EVI may have similar scales like dNBR
for interpretation. Via RBR, pixel values larger than 0.42 were not observed and the RdNBR index showed most of the pixel values
mainly within the range of 0.66 to 2.00, suggesting its tendency towards higher pixel values.

In the second panels of Fig 6 (a.2, b.2, and c.2), the histograms of the different burn severity indices are provided. The frequency
of dNBR, dNBR-EVI, and RBR exhibit nearly symmetric distribution shapes. While the histogram representing RdNBR is skewed
to the left, suggesting a bias towards higher pixel values. RBR histograms show a central tendency toward lower pixel values with
a narrow spread, highlighting this point that with lower range, RBR is less sensitive to burn severity variation. The histograms
demonstrating the distributions of dNBR and dNBR-EVI exhibit almost similar spread ranges and almost similar central tendencies
with pixel values larger than central tendency of RBR histogram. The RdNBR histograms display a widespread range and a central
tendency toward larger pixel values. Although RdNBR pixel values lower than -2 and greater than 2 are common, the number of
out-of-range RdNBR pixels is sufficiently low that they were not observable in the histograms. As a result, the x-axis was limited
to the range of -1 to 2.






**Figure 6.** Burn severity estimates via dNBR, dNBR-EVI, RBR, and RdNBR for three large fires (area >= 10k ha) as examples are presented in panels (a.1, b.1, and c.1). The thresholds applied are according EFFIS classification for dNBR (European Commission, 2018; Llorens et al., 2021). However, to facilitate the comparison of different severity indices, the same color classification was applied on all the indices. In the second panel, the histograms represent the distribution of different burn severity estimates via different indices.

**4 Discussion**

Portuguese Burn Severity Atlas includes estimates for 63 % of the total burned area between 1984 to 2022. The remaining 37 % of burned area without burn severity estimates were mainly due to the exclusion of fires <500 ha during 1984 to 2000 (25 %) and <100 ha for fires of 2001 onwards (10 %), and lack of satellite imagery to estimate burn severity (2 %). When considering only the fires with start and end date, only 3 % area of the burned area did not have burn severity estimates due to either lack of Landsat imagery with clear pixels within our sampling period or due to limitations of GEE performing heavier processing and stopping the code after the fourth iteration.

As it can be seen in Figure 2(b), 15 % of the dNBR pixel values are lower than 0.1. According to EFFIS (European Commission, 2018) burn severity discrete classification (European Commission, 2018; Llorens et al., 2021), dNBR pixel values less than 0.1 can be considered "Unburnt or Regrowth of Vegetation". This can be caused by either commission errors in the burned area mapping or due to the high regrowth potential of Portuguese vegetation cover, especially within the first month posterior to fire occurrence (Neves et al., 2023).

Results show that in almost 40 years, Portugal has on average experienced "High" burn severity. Mateus and Fernandes, (2014) detailed the following reasons behind the high severity fire regime in Portugal: 1) dominance of highly flammable *Eucalyptus globus* especially in Centro and Norte regions accounting for 77 % of all forest fires, 2) 90 % of all forest fires occurring within the months of June to September as the result of droughts, 3) dry conditions of dominant vegetation types due to seasonal weather patterns, 4) high productivity of understory plants specifically shrubs, 5) dominance of stand-replacing and crown fires, and most importantly 6) prioritizing fire suppression over prevention.

No visible trend was observed regarding the annual burned area extent over the studied years (Fig 3); however, the three largest fire seasons were in 2003, 2005, and 2017. Many fire research studies focused on these three years, not only because of their huge magnitude of burned extent, but also due to their different drivers (Beighley and Hyde, 2018) and consequences (Nitzsche et al., 2024). Oliveira et al., (2021) also found no trend regarding the burned extent over the years in Portugal while highlighting that 2017 was "the worst year" regarding the burned area. For these three years, more than 96 % of valid fires have burn severity estimates although imagery from different sensors were applied - Landsat-5 TM and Landsat- 8 OLI - due to different satellite availability dates (Table 1).

More reliable fire data since 2001 are available for Portugal as better means of data record technologies from various sources have been applied (Nunes et al., 2016). For recent years – 2013 onwards, higher percentage of burned area with severity estimates (on average ≈98 %) with no drastic variation were obtained, for which only images from Landsat- 8 OLI sensor were used. Hence, our omission error for fires posterior to 2001 are less than the ones prior to this year. Although both Landsat-5 TM and Landsat-8 OLI sensors have a similar temporal resolution, it was observed that via Landsat-8 OLI, more images with clear pixels –minimum reflectance contamination such as cloud, cirrus, shadow, and smoke – were available during the sampling period of ±110 days. This can be explained due to the difference in "daily acquisition image rate" of Landsat-8 OLI and Landsat-5 TM. Acquisition image rate is defined as the number of images acquired by each sensor on a daily basis. Landsat- 5 TM acquires 225 to 250 images





per day, which varies based on various factors such as the amount of sunlit land (Loveland and Dwyer, 2012). While Landsat-8

OLI acquires 725 images per day (Loveland and Irons, 2016). This point can be confirmed by the time series boxplot of fire-by-fire confidence (%) – average of SUITABILITY of pre- and post-fire images- showing smaller range variations with high average values (>80 %) for recent years (Fig 5). Higher confidence is the result of greater SUITABILITY value, which occurs when time lag decreases (Eq. (1)). Thus, in recent years, the time lags variation range should also be smaller which is confirmed by Fig B1 (mentioned in Sect 3.2).

Burn severity estimates were obtained from different Landsat sensors to ensure "spectral consistency" over the long study period (Manuel et al., 2024). However, the influence of different sensor characteristics in burn severity estimates cannot be discarded. No assessment in this regard was performed as the sensors availability dates do not overlap (Table 1). However, there are multiple studies which used burn severity estimates from different sensors and no incoherency has been reported (Singleton et al., 2019; Guindon et al., 2021; Mueller et al., 2020). Through the development of this atlas, bands from atmospherically corrected surface

reflectance images from Level 2, Collection 2, Tier 1 (Table 1) with the most similar wavelength from different Landsat sensors were used to minimized any possible inconsistency of the bands reflectance (Whitman et al., 2020). As stated by. Vogelmann et al., (2016), although there are minor changes to Landsat sensors, their spectral characteristics are still "reasonably comparable" and other factors such as smoke and haze are more influential to having impacts on spectral signals in comparison to differences in sensors' characteristics. While Poursanidis et al., (2015) claimed that Landsat- 8 OLI sensor provides more accurate results in

comparison to Landsat- 5 TM, though these were addressed to landcover mapping and not burn severity estimation.

We define RS images with "quality" as images with small time lags and preferably with little to no reflectance contaminations (e.g., cloud, cirrus, shadow, and smoke). Via the application of images with high quality, the most reliable burn severity estimates can be obtained (Miller et al., 2023; Dos Santos et al., 2020; Keeley et al., 2008). However, the number of high quality images over each individual fire perimeter is scarce, especially when considering the older sensors (Gao et al., 2006). To increase the

coverage of burn severity estimates, we considered an extensive sampling period (±110 days) and incorporated an "iteration process", improving the coverage of our atlas by 14 %. To the best of our knowledge, no other studies in literature have applied such a process before. However, mean compositing of different scenes to have burn severity or other types of atlases have been practiced (Parks et al., 2018; Whitman et al., 2020; Neves et al., 2023).

Due to our relatively long sampling period and as no specific scene acquisition row and/or path were determined when applying

Landsat imagery, the occurrence of seasonal variation and "mismatched phenology" between the pre- and post-fire images could have happened (Parks et al., 2018; Key, 2006; Storey et al., 2005; Lutes et al., 2006) as phenology is not constant throughout the year (Balata et al., 2022). On average, the worst pre- and post- fire time lags of this atlas were lower than ±50days for the first iterated burn severity estimations (Fig A6), thus, on average the seasonality influences have been minimized, although no further assessment in this regard was performed. More research related to seasonality and phenology analysis is necessary (Key, 2006;

Howe et al., 2022; Parks et al., 2018). In this atlas, we have calculated the offset values of burn severity indices of individual fires according to their corresponding formulas (Table 2). As indicated by Parks et al., (2018), offset value are accounted to differentiate between phenology variation between pre- and post-fire images. Hence, they can be applied to minimize the impacts caused by any possible "mismatched phenology" between pre- and post-fire images.

The definitions related to sampling periods are ambiguous. Our sampling period is ±110 days to avoid and minimize capturing

environmental and ecological responses. In many studies related to the "trend and evolution analysis of burn severity", such as the one performed by Dillon et al., (2006), the burn severity estimates calculated within 6 months or ±180 days from the ignition date were excluded. Dillon et al., (2006) classified this sampling period as "initial assessment". On the other hand, by definition



provided by Key, (2006), our sampling period is categorized as extended assessment. Thus, we have called our sampling period "rapid" to "extended assessment".

As burn severity estimates are highly time-sensitive, with the increase of time lag, the accuracy of RS estimates decreases and degrades as environmental responses are cumulated over fire impacts (Key, 2006). To have an understanding over this degradation, we modeled the variation of dNBR, as the standard burn severity index, caused by the increase of time lag (Fig 4). The dNBR tends to increase with the increase of the pre-fire time lag, which leads to an overestimation of burn severity. In Portugal, as in all Mediterranean-climate areas, vegetation vigor is lower before fire occurrence as a result of high temperatures and solar radiation,

and low water availability (Verbyla et al., 2008; Chu and Guo, 2014; Pascolini-Campbell et al., 2022; Fernández-Guisuraga et al., 2023b). Generally, the pre-NBR tends to decrease as the fire season approaches (Alonso-González and Fernández-García, 2021). Thus, with the increase of pre-fire time lag, burn severity is overestimated as the amount of vegetation considered burned is overestimated. The dNBR tends to decrease with the increase of the post-fire time lag, which leads to an underestimation of burn severity. This can result from the fire scars becoming less visible due to environmental and ecological responses such as

resprouting, especially in Portugal where vegetation tends to regrow within the first month after the fire (Neves et al., 2023) and/or via ashes being washed off by rain and wind (Key, 2006; González-Pelayo et al., 2023, 2024).

Burn severity estimates via different indices reveal varying degrees of post-fire impacts across the landscape (Parks et al., 2014; Miller and Thode, 2006; Fernández-García et al., 2018). Combining information from multiple indices could offer a more comprehensive understanding of burn severity. Thus, in this atlas, we provided burn severity estimates using four dNBR-derived

indices. For instance, as can be observed from the provided examples in Fig 6, for the same fires, each of the used indices can contribute to better interpretation of burn severity. In summary, dNBR-EVI can be applied to distinguish between "regrowth" and "unburned" classes. As RdNBR is more sensitive to vegetation type than dNBR, its tendency to higher pixel values can be interpreted as low fuel load of vegetation cover (Miller and Thode, 2006) of the chosen fires. However, more in-depth analysis is needed to confirm this point. Within these examples, RBR was observed to be more prone to signal saturation. Signal saturation

of dNBR-derived indices is defined as incapability of indices in measuring very high burn severity with its value reaching a certain point where they no longer are capable of discerning subtle differences in burn severity (Veraverbeke et al., 2012). Thus, caution should be taken when interpreting the dNBR-derived burn severity maps, as they are subjected to signal saturation (Fernández-García et al., 2018; Santis et al., 2010; Fernández-Guisuraga et al., 2023a), especially RBR. As stated by Fernández-Guisuraga et al., (2023a) any interpretation of RS burn severity estimates must be accompanied and confirmed with ground truth data specifically

for relative forms like RdNBR and RBR (Miller and Thode, 2006; Parks et al., 2014; Cansler and McKenzie, 2012).

In this atlas, no data regarding the ground burn severity assessment is neither included nor analyzed. The provision of means of interpretation of burn severity degrees or classes of our maps is not within the scope of this study. Thus, no classification thresholds for any of burn severity indices is proposed and means of interpretation of burn severity must only align by user's objectives. However, as an example of means towards interpretation of burn severity, in this study, we have provided the thresholds assigned

by EFFIS. This point must be highlighted that the thresholds by EFFIS are assigned only for dNBR index and not for other indices. In other studies, based on the correlations between CBI and dNBR, RdNBR and RBR, different thresholds have been introduced, however, only for specific parts of USA (Alonso-González and Fernández-García, 2021; Parks et al., 2018). Thus, to the best of our knowledge, aside from EFFIS's dNBR classification, no classification regarding other burn severity indices have been proposed for Mediterranean regions, and especially for Portugal. Hence, our maps have all been presented in their continuous raw forms and

no classifications have been applied to them.

Portuguese Burn Severity Atlas can be used as the foundation of many future research projects and presents numerous research opportunities. With 97 % of burn severity estimates of valid fires from 1984 to 2022, we can confidently state that characterization



of long-term burn severity patterns (Singleton et al., 2019; Gale and Cary, 2022), burn severity heterogeneity related analyses (Lutz et al., 2011; Buonanduci et al., 2023), analysis of isolating burn severity environmental and climate variables (Miller et al., 2009),

post-fire recovery studies (Oliveira et al., 2011; Alonso-González and Fernández-García, 2021; Whitman et al., 2020) and fire consequences studies (Wells et al., 2021; Petratou et al., 2023; Amerh et al., 2022; Vieira et al., 2023; Singh et al., 2022) can be conducted. Moreover, as suggestions for future research and via this atlas, we can highlight the gap of knowledge in interpretation of RS burn severity estimates especially the relative forms in the Mediterranean regions. Secondly, the most influential burn severity drives, both environmental and climatic, are required to be distinguished and accordingly, more informed pre- and post-

fire management plans should be formed to minimize future fire impacts in Portugal. Last but not least, we encourage the execution of trend analysis of burn severity evolution to assess whether the burn severity in the mainland Portugal has changed over the years.

## 5 Data availability

The maps of Portuguese Burn Severity Atlas can be accessed at https://doi.org/10.5281/zenodo.12773611 (Jahanianfard et al.,

2024). The annual maps are provided in subfolders entitled as the corresponding year. Annual fire perimeters (shapefile) along with a table with details on pair of images used for each iteration, the confidence, and the offset values of burn severity indices are also stored within these subfolders.

## 6 Code availability

The GEE code can be accessed at https://code.earthengine.google.com/b23081d3643bc46585d73f893b9efdab?noload=true, with

the fire data already shared as assets, however, it is necessary to have a GEE account to access the code. The code can also be accessed    at    https://github.com/DinaJahanianfard/Portuguese-Burn-Severity-Atlas_v2/commit/7aee76ea5b3df0db8cd047a4b8cf6624bf965d50 with no account nor registration needed.

## 7 Conclusion

In this study, a comprehensive Portuguese Burn Severity Atlas was developed spanning from 1984 to 2022, derived from Landsat

satellite imagery (30 m resolution), with the exception of the year 2012, for which MODIS imagery was used. This atlas contains burn severity estimates for 63 % of the 4.92 million ha burned between 1984 and 2022, and 97 % of the valid fires, with a total burn area of 3.19 million ha. The atlas illustrates that Portugal has on average experienced "high" burn severity over the study period.

Through "iteration process", we expanded the coverage of the atlas, providing burn severity estimates for an additional 14 % of

the valid fire area, totaling 0.44 million ha. Furthermore, we developed a semi-automated code in Google Earth Engine (GEE) that can be easily updated and modified by users to generate burn severity estimates for any region worldwide, using any desired sampling period, with only the requirement of fire data.

Our findings regarding the relationship between dNBR and time lag indicate that increasing pre-fire time lags result in overestimation of burn severity, while increasing post-fire time lags lead to its underestimation.

Analysis of burn severity estimates using different indices reveals distinctive characteristics. Notably, the dNBR-EVI index demonstrates potential for distinguishing between "unburned" and "regrowth" classes. RdNBR tends to indicate higher burn severity, while RBR shows a tendency toward signal saturation compared to other indices. However, further investigation is

necessary to validate these findings, particularly in the context of defining thresholds for interpreting burn severity classes in Mediterranean regions.

Ultimately, the burn severity maps provided in this atlas offer numerous opportunities for research across various disciplines. They enable investigations into burn severity heterogeneity, trend analysis, environmental and climatic drivers of burn severity, as well as studies related to air and water quality and soil erosion.

**Appendix A - Supporting material for the methods**





**Figure A1.** Flowchart representing the iteration scenarios. PreIC and PostIC represent the pre- and post-fire image collections in
GEE and their size or the number of images within each IC is shown (e.g., PreIC <= 3 meaning the number of images within this IC is less or equal to 3). All the ICs are sorted from images with the highest SUITABILITY to the lowest and the first image, shown as e.g., PostIC(0), is the first image within the sorted post-fire IC with the highest SUITABILITY.





**Figure A2.** Scenario: not_ intersected - The dNBR estimation for the whole area of this fire (area = 4029 ha, year= 2003, ID= 2904), was obtained through 3 iterations. With changing both the pre- and post-fire images (the first iteration with highest SUITABILITY pre-and post-fire images and covering the area of 3730 ha, second iteration with changing the pre-fire image and covering the area of 160 ha, and third iteration with changing the post-fire image and covering the area of 165 ha). Panel (i) shows the iteration steps while panel (ii) shows the final dNBR and confidence maps of this fire.

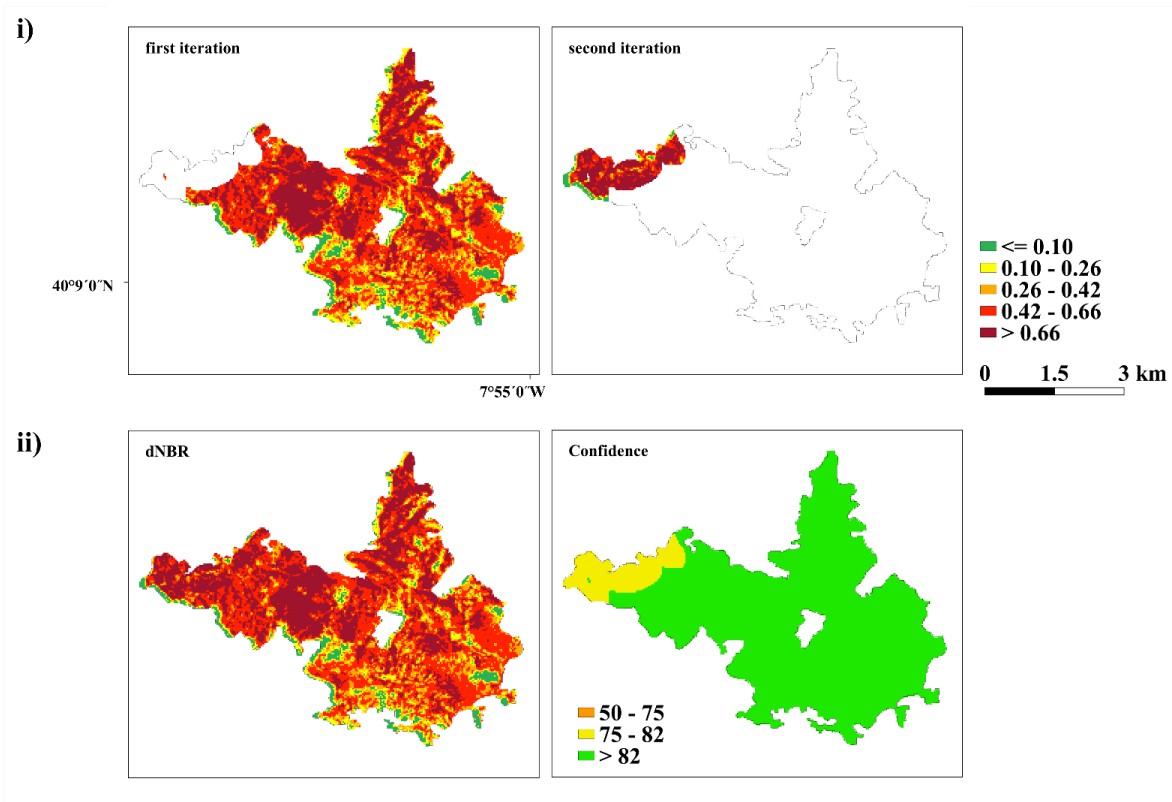

**Figure A3.** Scenario: intersected - The dNBR estimation for the whole area of this fire (area = 2071 ha, year= 2003, ID= 2180), was obtained through 2 iterations. (the first iteration= pre- and post-fire images with the highest SUITABILITY and covering the area of 1903 ha, second iteration= second highest pre-and post-fire images and covering the area of 168 ha). Panel (i) shows the iteration steps while panel (ii) shows the final dNBR and confidence maps of this fire.

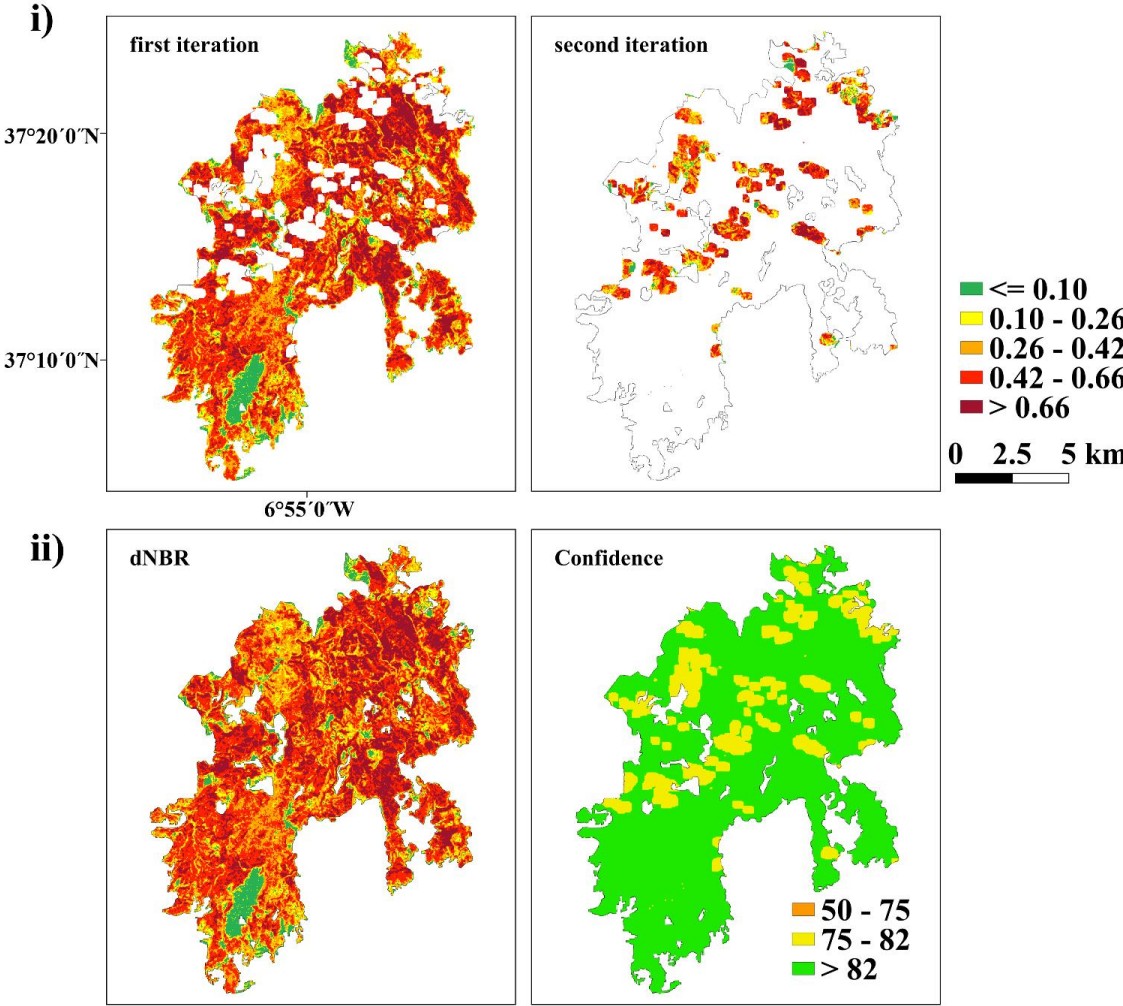


**Figure A4.** Scenario: Pre - The dNBR estimation for the whole area of this fire (area = 13770 ha, year=2003, ID= 2933), was obtained through 2 iterations. The first iteration with the highest SUITABILITY pre- and post-fire images covering the area of 11631 ha, second iteration with the second highest SUITABILITY pre-fire and the highest SUITABILITY post-fire image and covering the area of 2143 ha). Panel (i) shows the iteration steps while panel (ii) shows the final dNBR and confidence maps of

this fire.






**Figure A 5.** Scenario: Post - The dNBR estimation for the whole area of this fire (area = 8571 ha, year= 2003, and ID=2416), was obtained through 4 iterations. The first iteration= highest SUITABILITY pre- and post-fire images covering the area of 5568 ha, second iteration= the highest SUITABILITY pre-fire and the second highest SUITABILITY post-fire image covering area of 2626



ha, third iteration= the highest SUITABILITY pre-fire and the third highest SUITABILITY post-fire image **covering area of 30 ha, and** fourth iteration= with the highest SUITABILITY pre-fire image and fourth highest SUITABILITY post-fire image covering the area of 347 ha). Panel (i) shows the iteration steps while panel (ii) shows the final dNBR and confidence maps of this fire.

**Appendix B - Supporting material for the results**

**Table B1.** Annual table summarizing all data regarding, total area of burned areas, total area of valid burned areas, total area with dNBR estimation, and number of all and valid burned areas, and percentages of areas with dNBR estimations.

| Year | Sum of area of all fires | Sum of area of valid fires | Sum of area with reference dNBR | Sum of area with iterated dNBR | Sum of area with dNBR estimation | Area of valid to all fires (%) | Area with dNBR to all fires (%) | Area with dNBR to valid fires (%) | Number all fires | Number valid fires | Number of valid to all fires (%) | Sensor |
|---|---|---|---|---|---|---|---|---|---|---|---|---|
| 1984 | 116809 | 19244 | 16222 | 2979 | 19201 | 16 | 16 | 100 | 1845 | 27 | 1.5 | Landsat5 |
| 1985 | 289960 | 97268 | 83441 | 11792 | 95233 | 34 | 33 | 98 | 2631 | 118 | 4.5 | Landsat5 |
| 1986 | 112168 | 46885 | 34948 | 9309 | 44257 | 42 | 39 | 94 | 1536 | 33 | 2.1 | Landsat5 |
| 1987 | 137761 | 54843 | 29874 | 17528 | 47286 | 40 | 34 | 86 | 1556 | 46 | 3.0 | Landsat5 |
| 1988 | 31321 | 2346 | 2058 | 285 | 2343 | 7 | 7 | 100 | 656 | 4 | 0.6 | Landsat5 |
| 1989 | 204028 | 79036 | 58836 | 19765 | 78601 | 39 | 39 | 99 | 2242 | 88 | 3.9 | Landsat5 |
| 1990 | 105858 | 47867 | 44858 | 2897 | 47755 | 45 | 45 | 100 | 1416 | 40 | 2.8 | Landsat5 |
| 1991 | 182190 | 117960 | 88125 | 17976 | 106101 | 65 | 58 | 90 | 880 | 60 | 6.8 | Landsat5 |
| 1992 | 34222 | 15123 | 14009 | 1074 | 15083 | 44 | 44 | 100 | 230 | 8 | 3.5 | Landsat5 |
| 1993 | 40239 | 23400 | 20124 | 3177 | 23301 | 58 | 58 | 100 | 141 | 18 | 12.8 | Landsat5 |
| 1994 | 72005 | 24018 | 17207 | 6040 | 23247 | 33 | 32 | 97 | 623 | 31 | 5.0 | Landsat5 |
| 1995 | 134474 | 68551 | 57492 | 9283 | 66775 | 51 | 50 | 97 | 1750 | 46 | 2.6 | Landsat5 |
| 1996 | 92938 | 31039 | 24201 | 6718 | 30919 | 33 | 33 | 100 | 1477 | 32 | 2.2 | Landsat5 |
| 1997 | 21265 | 0 | 0 | 0 | 0 | 0 | 0 | 0 | 755 | 0 | 0.0 | - |
| 1998 | 216147 | 88365 | 73861 | 14302 | 88163 | 41 | 41 | 100 | 1837 | 80 | 4.4 | Landsat5 |
| 1999 | 67167 | 13888 | 10393 | 3466 | 13859 | 21 | 21 | 100 | 1462 | 14 | 1.0 | Landsat5 |
| 2000 | 143291 | 59079 | 50793 | 4352 | 55145 | 41 | 38 | 93 | 1733 | 54 | 3.1 | Landsat5 |
| 2001 | 95936 | 43981 | 36140 | 7195 | 43335 | 46 | 45 | 99 | 1349 | 93 | 6.9 | Landsat5 |
| 2002 | 130440 | 74557 | 61056 | 12618 | 73642 | 57 | 56 | 99 | 1422 | 163 | 11.5 | Landsat7 |
| 2003 | 420985 | 386257 | 312374 | 61424 | 373798 | 92 | 89 | 97 | 936 | 188 | 20.1 | Landsat5 |
| 2004 | 119762 | 93380 | 75500 | 16612 | 92112 | 78 | 77 | 99 | 634 | 97 | 15.3 | Landsat5 |
| 2005 | 334934 | 295681 | 236064 | 52057 | 288121 | 88 | 86 | 97 | 693 | 326 | 47.0 | Landsat5 |
| 2006 | 74141 | 53183 | 35131 | 4861 | 39992 | 72 | 54 | 75 | 570 | 96 | 16.8 | Landsat5 |
| 2007 | 35093 | 15897 | 10306 | 622 | 10928 | 45 | 31 | 69 | 379 | 44 | 11.6 | Landsat5 |
| 2008 | 9976 | 3774 | 3453 | 320 | 3773 | 38 | 38 | 100 | 212 | 11 | 5.2 | Landsat5 |
| 2009 | 74796 | 45085 | 43396 | 1378 | 44774 | 60 | 60 | 99 | 518 | 94 | 18.1 | Landsat5 |
| 2010 | 129767 | 102403 | 84433 | 17132 | 101565 | 79 | 78 | 99 | 710 | 139 | 19.6 | Landsat5 |
| 2011 | 76993 | 50788 | 40386 | 528 | 40914 | 66 | 53 | 81 | 710 | 120 | 16.9 | Landsat5 |
| 2012 | 99527 | 81045 | 81100 | 0 | 81100 | 81 | 81 | 100 | 560 | 112 | 20.0 | MODIS |
| 2013 | 147692 | 115391 | 102105 | 12518 | 114623 | 78 | 78 | 99 | 849 | 170 | 20.0 | Landsat8 |
| 2014 | 17174 | 10233 | 8068 | 1996 | 10064 | 60 | 59 | 98 | 186 | 19 | 10.2 | Landsat8 |
| 2015 | 58272 | 42683 | 40239 | 2216 | 42455 | 73 | 73 | 99 | 428 | 82 | 19.2 | Landsat8 |
| 2016 | 163881 | 145884 | 123440 | 21073 | 144513 | 89 | 88 | 99 | 645 | 185 | 28.7 | Landsat8 |
| 2017 | 562557 | 543673 | 488664 | 53793 | 542457 | 97 | 96 | 100 | 846 | 264 | 31.2 | Landsat8 |
| 2018 | 39912 | 33838 | 33503 | 234 | 33737 | 85 | 85 | 100 | 240 | 25 | 10.4 | Landsat8 |
| 2019 | 39073 | 28243 | 23736 | 4060 | 27796 | 72 | 71 | 98 | 440 | 64 | 14.5 | Landsat8 |
| 2020 | 71143 | 62326 | 55650 | 5127 | 60777 | 88 | 85 | 98 | 359 | 72 | 20.1 | Landsat8 |
| 2021 | 25413 | 15511 | 14400 | 937 | 15337 | 61 | 60 | 99 | 379 | 35 | 9.2 | Landsat8 |
| 2022 | 193407 | 164750 | 129803 | 33073 | 162876 | 85 | 84 | 99 | 2882 | 142 | 4.9 | Landsat8 |
| **total** | **4922717** | **3197475** | **2665389** | **440717** | **3105958** | **65** | **63** | **97** | **38717** | **3240** | **8.4** | - |
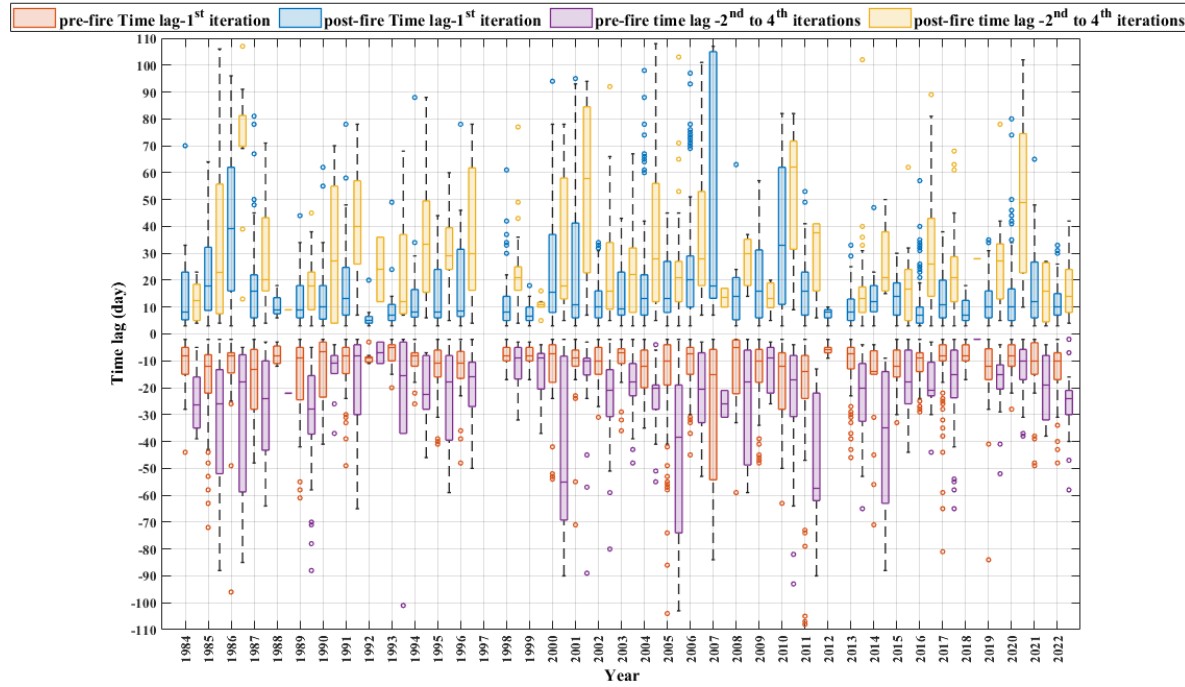


**Figure B1.** The annual variability of pre- and post-fire time lags in unit of days are presented, both from the first and second to fourth iterations. Negative values represent pre- fire time lags and positive ones show post-fire time lags. No data is presented for 1997 (no valid fire data).

**Authors contribution**

Jahanianfard. D and Benali. A designed the study. Benali. A and Parente. J provided the initial corrected fire data. Jahanianfard. D applied secondary corrections on the fire data, developed the code in GEE, developed and processed the severity maps and wrote the first draft of the manuscript. Benali. A, Gonzalez-Pelayo. O and Parente. J reviewed and provided comments on the first draft. Jahanianfard. D applied the comments and finalized the manuscript.

**Competing of interests**


The authors declare that they have no conflict of interest.



**Disclaimer**

The authors declare no disclaimer.

**Acknowledgement**

We thank Bruno Aparício, Chiara Bruni, and Beatriz Lourenço for their contribution in applying primary correction on fire data for fires of 2001 to 2022. We acknowledge the Portuguese Foundation for Science and Technology (FCT) for the funding of FRISCO (PCIF/MPG/0044/2018), the project which the initial idea of burn severity atlas for Portugal was first introduced. Special thanks to FCT for the financial support of Jahanianfard. D through the national PhD grant (2021.08094.BD). Thanks are due to

FCT/MCTES for the financial support for CESAM (UIDP/50017/2020+UIDB/50017/2020 + LA/P/0094/2020) and to the University of Aveiro through the assistant research contract of González-Pelayo. O (CDLCTTRI-97-ARH/2018. REF. 190-97-ARH/2018).

**Financial support**

Jahanianfard. D is supported by the Portuguese Foundation for Science and Technology (FCT-Fundação para a Ciência e

Tecnologia) with a PhD grant reference (https://doi.org/10.54499/2021.08094.BD).

**Review statement**

This part will be completed.

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
