# Peer review of "Multidecadal satellite-derived Portuguese Burn Severity Atlas (1984–2022)"

_Earth System Science Data, 2024_

## Author Comment (AC1)

**Reviewer #1**

The present study is concerned with the multidecadal satellite-derived Portuguese Burn Severity Atlas. The authors offer valuable insights into the topic. The authors present the following observations for consideration.

We thank the reviewer for their feedback and comments.

1. It is my contention that the role of small fires in the context of wildfire management is a significant yet understudied area. In contrast, the current study has a narrow focus on large fires (>100 ha).

We thank the reviewer for their insight. We focused on large fires (>=100 ha) because these correspond to the majority of area burned in Portugal. As stated by Fernandes et al 2009 and records from Divisão de Defesa da Floresta Contra Incêndios (DGRF), large fires correspond to only 1% of all fires but correspond to 75 % of the total burned area (mentioned in the manuscript in sub-section "2.1 Fire Data" lines 129 and 131). Concurrently, in our study, 37 % of all the recorded fires (1984 -2022) had no burn severity estimates:

- 25 % corresponded to pre-2000 fires smaller than 500 ha and/or with no recorded dates
- 10 % corresponded to post-2001 fires smaller than 100 ha and/or with no recorded dates
- 2 % corresponded to lack of satellite imagery.

These aspects are mentioned in the section "4 Discussion" lines 399 to 404.

In addition, as there are a lot of small fires throughout the years in Portugal, accurate fire data has not been recorded for all fires. This fact is unfortunately worse for historical fires as we were not even able to find dates for pre-2000 fires between 100 ha and 500 ha and due to the large difference between fire and image acquisition date (introduced as time lag throughout the manuscript), we were forced to only consider fires large than 500 ha for this duration (this point is mentioned in the sub-section "2.1 Fire data" in lines 149 and 150).

As our methodology can be applied to smaller fires (<100 ha), we consider our focus only on large fires as a limitation of our study, which in future studies can be explored. Hence, the need for burn severity analysis of smaller fires is mentioned in the section "4 Discussion" in lines 404 to 406. The added point in the revised version is as follows:

Within this study, the focus was on large fires (>=100 ha), however, this can be considered as a limitation of our atlas. As our developed methodology can be applied to smaller fires (<100 ha), the exploration of burn severity of small fires can be a research opportunity for future studies.

2. It is anticipated that the fire atlas will have an identical resolution, resulting in comparable outcomes at either 30 or 500 meters. It is recommended that the ALI sensor be used for the 2012 data set.

We thank the reviewer for their suggestion. We agree that it would have been ideal for the atlas to have identical resolution (preferably 30 m) throughout the years. Aside from the Landsat series, images from Earth Observation-1 Advanced Land Imager (hereafter as EO-1 ALI) would have been the best option considering the spatial resolution. However, we have not created the atlas by manually gathering and processing the images, instead we have used "Google Earth Engine" and created a semi-automated process to gather, select, process, and calculate the burn severity estimates. Unfortunately, the EO-1 ALI dataset is not available on Google Earth Engine.

Moreover, although EO-1 ALI sensor had the capability of having global coverage, it was an experimental satellite and mission-based, hence, it did not have continuous and frequent imagining with global coverage

like Landsat series of sensors. Thus, the number of images available over Portugal throughout the years is scarce with total number of available images of 52 which all were captured before 2012 (only from 2002 to 2011 (https://earthexplorer.usgs.gov/).

As 112 large fires occurred in 2012 with the approximate extent of 81000 ha, there are not any available EO-1 ALI images. Nonetheless, manually gathering and processing images would be time consuming and is beyond the scope of the work. Hence, we decided to apply images from Terra aboard Moderate Resolution Imaging Spectroradiometer (MODIS) with a resolution of 500m for 2012.

This point has been added in the section "4 Discussion" in lines 440 to 451 as follows:

This objective was achieved for all years except for 2012 for which images from MODIS Terra were applied which has caused Portuguese burn severity atlas not to have consistent spatial resolution throughout all the years. This aspect is a limitation of the atlas. According to Landsat sensors' availability dates (Table1), no images from this family of sensors were available for this year aside from Landsat-7 ETM+ with SLC failure. One possible alternative would be to use images from Earth Observation-1 Advanced Land Imager (hereafter EO-1 ALI) available from November 2000 to March 2017 with a spatial resolution of 30 m (Chander et al., 2009). Although EO-1 had the capability of imagining with global coverage, it was an experimental and mission-based satellite (Hoang and Koike, 2018) and via accessing the website (EarthExplorer, 2025), it was observed that there are only 52 images available form this sensor with coverage of Portugal's mainland, and all were captured before 2012 (from 2002 to 2011). Hence, to have temporal consistency, images from MODIS Terra were applied for 2012. According to by Alonso-González and Fernández-García, 2021, the burn severity estimates obtained by Landsat and MODIS are comparable despite the big difference in their resolution and to evaluate their comparability, their developed methods can be applied (Alonso-González and Fernández-García, 2021).

3. It is my contention that a burn severity mapping system based solely on these indices with a fixed threshold will not yield optimal results. The resulting burn severity map is an inaccurate representation of the landscape due to the influence of environmental conditions, the diversity of objects, and the impact of climate.

We thank the reviewer for this comment. All our maps are presented in their continuous forms and no fixed threshold and classification have been applied on them. We have specified this point in the section "4 Discussion" lines 530 and 531.

We only presented the thresholds assigned by European Forest Fire Information Service (hereafter EFFIS) as an example of interpretation means to facilitate interpreting the maps by users if needed. However, means of interpretation must only be aligned by the users' objectives. We have emphasized this point in the section "4 Discussion" in lines 516 to 518. Moreover, we have specified in the section "4 Discussion" lines 511 to 512, that any means of interpretation must be accompanied by field observations and users are required to take "caution" using any predefined means of interpretation. The reason for this caution is also added as follows in in the section "4 Discussion" lines 512 to 515:

This is crucial because burn severity often varies across vertical strata, and satellite-derived reflections are differently sensitive to impacts at each layer. Aggregating these impacts into a single metric can obscure important ecological details (Fernández-Guisuraga et al., 2023a; Miller and Thode, 2006; Parks et al., 2014; Cansler and McKenzie, 2012).

This point must also be highlighted that the fixed thresholds by EFFIS are provided based on the comparison that they made between dNBR pixel values and ground burn severity assessment in the Mediterranean regions, considering the dominant environmental and climate conditions within these regions.

This point will be added in the section "4 Discussion" in lines 518 to 522.

The added point is as follows:

As an example of means towards interpretation of burn severity, in this study, the thresholds assigned by EFFIS are mentioned. The thresholds of EFFIS are assigned only for dNBR index and not for other indices and they are obtained from the comparison between dNBR pixel values and ground burn severity estimates considering the dominant environmental and climatic conditions within the Mediterranean regions (Llorens et al., 2021).

4. It is anticipated that the reliability of the atlas dataset will be enhanced by the incorporation of a more robust validation dataset.

We thank the reviewer for their comment. We have not used any "validation dataset" as we emphasized that no ground burn severity assessment was performed and no data in this regard were included nor analyzed (mentioned in the section "4 Discussion" in line 516). All these maps are estimations of burn severity and to assess their validity, they must be accompanied by ground burn severity assessment. The scope of this work does not include any validation.

There are multiple studies in which the validation of different burn severity indices used in this atlas have been performed and they are mainly focused on different parts of the USA as mentioned in the section "4 Discussion" lines 522 to 534.

Example of other studies with their focus on the Mediterranean regions will be added in the section "4 Discussion" lines 530 to 538. However, these studies are isolated and limited and hence, they cannot be incorporated on large scale, which in our case is the mainland of Portugal (e.g., the study performed by Fernández-García et al (2022) considering the total number of 23 fires among which only 4 fires were located in Portugal). While lack of validation is a limitation in our study, as a suggestion for future studies, this point will be also mentioned that the conduction of such analysis is encouraged. These added points will be in the section "4 Discussion" lines 524 to 532.

The added points are as follows:

In Spain and specifically in Valencia province, by comparing CBI and dNBR, RdNBR, and RBR, the classification thresholds for interpretation of these indices have been introduced (Botella-Martínez and Fernández-Manso, 2017), which were furthered utilized for interpretation of burn severity estimates by ICNF for fires which burned in Monchique and Portimão in Portugal (Anon, 2021). Moreover, there is a study conducted by Fernández-García et al (2022) considering the total number of 23 fires among which only four fires were located in Portugal (Fernández-García et al., 2022). Although there are studies comparing burn severity observations with estimates, they are isolated and limited and hence, they cannot be incorporated on large scale, which in our case is the mainland of Portugal. Hence, our maps have all been presented in their continuous raw forms and no classifications have been applied to them. We acknowledge the lack of validation as a limitation of our atlas and we encourage conduction of future studies to validate the burn severity estimates of Portuguese Burn Severity Atlas.

5. It would be beneficial to examine the influence of seasonality and the climate dataset on the severity map.

We thank the reviewer for their suggestion. The analysis of burn severity drivers including climate variables was not within the scope of this study, while strongly encouraged by us to be performed in future studies. Our main goal is to provide users with burn severity estimates as a necessary foundation enabling future studies to perform all sorts of analysis specifically on the correlation of burn severity and the drivers (both top-down and bottom-up).

"Seasonality" as a burn severity driver was not explored as it was not within the scope of this study. However, the influence of "seasonality" on the quality of burn severity estimates, has been briefly explored as we have pointed out its role for instance by causing "mismatched phenology" between pre- and post-fire images (mentioned in the section "4 Discussion" within lines 470 to 481), while it was not quantified.

Instead, we focused on the variation of the time lag and modelled it. Moreover, we tried to minimize the influence of the seasonality by capping the image acquisition sampling period, i.e., the duration to acquire satellite images, to ±120 days. By the analysis of the time lag variation from the first to fourth iterations, it was observed that the time lag on average does not exceed ±50 days (the corresponding result is presented in Figure B1 in the Appendix B section). However, the difference in seasonality may still have occurred and to minimize this impact, we have also calculated the burn severity offset values for all the indices and if the users are concerned with the seasonality variation, they can apply the offset values. Via the calculated offset values and comparing them to the time lag, the influence of seasonality can be also quantified. As the conduction of such analysis is encouraged by us, this point is added to "Discussion" section in lines 479 to 481. The added point is as follows:

Moreover, as suggestion for future studies, the lack of knowledge in the degree of seasonality influence on quality of burn severity estimates can be highlighted that via the offset values provided in the atlas and comparison between them and time lag such analysis can be conducted.

6. It would be advantageous to include the severity of different objects and trees during the fire period.

We thank the reviewer for their suggestion. Within this atlas, we applied images from the Landsat series of sensors with a resolution of 30 m. Hence, each pixel value corresponds to average reflectance of an area equivalent to 0.09 ha. Hence, distinguishment of burn severity of different land cover species and degree of burn severity of trees such as "crown damage" or "complete crown consumption" or in other words burn severity heterogeneity of different land cover types is not possible as this is a consistent limitation in capabilities of current sensors.

**Minor comments:**

1. It is recommended that all figures be enhanced in terms of quality.

We thank the reviewer for their comment. We will update the manuscript with higher quality figures. Moreover, in the revised version of the manuscript, the legend of Figure 5 was changed to "1st iteration (representing the blue boxplots).

2. It would be helpful to include a map indicating the frequency of burn for different areas.

We thank the reviewer for their suggestions. A figure is added to the Appendix A as Figure A1 to show frequency of fires – both all and valid fires- within the parishes and districts of Portugal. Reference to this figure was added in the sub-section "2.1 Fire data" in lines 162 to 165. The added point is as follows:
To have an overview on the frequency of fires within the mainland of Portugal, FigA1 is provided in the Appendix. The frequencies of fires in Northern and Central regions are higher in comparison to other regions based on level 3 "Nomenclature of Territorial Units for Statistics" (presented as NUTS3) frontiers classification (Registo Nacional de Dados Geográficos - Direção-Geral do Território (DGT), 2024; Meneses et al., 2018).

3. It would be beneficial to analyze the impact of the threshold value for different sensors on the same fire.

We thank the reviewer for their suggestion. As Landsat sensors' availability dates do not overlap (Table 1), it is not possible to have burn severity estimates for one fire via images acquired from different sensors. Hence, although this analysis would have been indeed beneficial, its performance is not possible.

[Figure]

Figure A1 – Fire frequency of fires within parishes and districts of Portugal are presented. In panel i) and ii) of this figure, the fire frequency of all fires burned throughout the years regardless of their size and with and/or without dates recorded are presented within "Nomenclature of Territorial Units for Statistics" level 2 (NUTS2) demonstrating Portugal's districts and Portugal's parishes, respectively. Panels iii) and iv) demonstrate the fire frequency of valid fires, i.e., fires larger than 500 ha for pre-2000 and larger than 100 ha for post-2001 with start and dates recorded, within Portugal's districts and parishes, respectively.

**Reviewer #2**

The data article provides a new dataset to potentially evaluate fire severity in Portugal in the period 1984-2022. This new dataset is a contribution to the analysis of fire severity in Portugal in the last decades. However, there are a series of limitations in the dataset that must be addressed prior to its publication:

The authors select fires larger than 500 ha for the period 1984-2000 to assess fire severity and fires larger than 100 ha for the remaining period, until 2022. The authors must justify the selection of these fire sizes and express in the methodology the percentage that these fires represent for Portugal in terms of number of fires as compared to the total number of fires in the country, and in terms of burnt area, as compared to the total burnt area in the country for all the years of the analysis. Fires larger than 100 ha are probably below 1% of the total number of fires, although they may have a larger contribution in terms of burnt area, maybe about 30-40%? These basic statistics are needed to know how representative the fire severity dataset is for Portugal.

On the basis of the above and the availability of satellite imagery from Landsat sensors, the authors must justify the selection of the 500 ha and 100 ha thresholds. In principle and given that the imagery is at 30 m spatial resolution, there is no obvious justification for the selection of these thresholds. Please, explain also why MODIS was used for 2012, which was a critical year of wildfires in Portugal and the potential inter-calibration (comparability) of the fire severity series for the rest of the years with the data obtained for 2012.

We thank the reviewer for their feedback.

As stated by Divisão de Defesa da Floresta Contra Incêndios (DGRF), (2006) and Fernandes, (2009), historically in Portugal, fires equal to and larger than 100ha correspond to 1% of numbers of fires while accounting for 75% of total burned extent. This point is mentioned in the subsection "2.1 Fire data" in lines 129 and 131. Hence, we set our fire size equal to or larger than 100ha. However, aside from the fire size, it was important that the fires considered for burn severity estimation had their start and end dates recorded as burn severity estimates without knowing dates will not make sense and also as our main objective was to provide the burn severity estimates for fires with the lowest time lag, i.e., minimum difference in unit of days between fire dates and the image acquisition dates. Unfortunately, for the period of 1984 to 2000, due to lack of accurate fire data, we were unable to find the dates for fires between 100ha and 500ha and the majority of fires (98.6%) within this time period with their dates recorded were equal to or larger than 500ha. So, for the period of 1984 to 2000, mainly fires equal to or larger than 500ha have their burn severity estimates provided in our atlas. This point is mentioned in the sub-section "2.1 Fire data" in lines 146 and 147. We called this subset of fires "valid fires" which correspond to fires with start and end dates recorded and their size equal to and larger than 500 ha for the duration of 1984 to 2000 and equal to and larger than

100ha for the duration of 2001 to 2022. We have provided such statistics within the "methodology" section of our manuscript in the sub-section 2.1 Fire data in lines 158 and 162. Moreover, as stated by Alvares et al., 2024, fires >=500 ha, in Spain with similar environmental and climatic conditions to Portugal, are more likely to reflect significance of burn severity. Hence, for conducting any future burn severity analysis, which is highly encouraged by us, this size threshold is not a limitation.

Regarding the selection of MODIS imagery for 2012, it was only due to the fact that for the year 2012 only Landsat sensor available (according to Landsat availability dates summarized in Table 1) is Landsat-7 ETM+ imagery. However, as mentioned in the sub-section "2.2 RS imagery: access and processing" in lines 176 to 177, this sensor suffered a technical failure in its scan line corrector (SLC) in May 2003 with multiple gaps within its imagery since then. Our explanatory analysis showed that for 2012 if we have used Landsat-7 ETM+ imagery, because of multiple gaps, we would have lost 52% of area burned in 2012 with no burn severity estimation. Hence to have "temporal coherency", for this year, we provided the burn severity estimates with imagery from MODIS.

Regarding "potential inter-calibration (comparability)" of burn severity estimates of 2012 with spatial resolution of 500m with burn severity estimates of other years with spatial resolution of 30m, according to Alonso-González, E and Fernández-García, V (2021), the burn severity estimates from Landsat and MODIS are comparable and their method of evaluating comparability can be applied. This point is added to "Discussion" section within the lines 449 to 451. The added point is:

According to by Alonso-González and Fernández-García, 2021, the burn severity estimates obtained by Landsat and MODIS are comparable despite the big difference in their resolution and to evaluate their comparability, their developed methods can be applied (Alonso-González and Fernández-García, 2021).

The statement on the use of the DNBR-EVI regarding the potential use of this index based on the analysis of 3 fires, only, should be excluded from the article as the representativity of this analysis is not acceptable by any scientific standards. The authors are encouraged to continue the testing of this index with a sufficiently large/representative dataset.

We thank the reviewers for their feedback. This point is removed in the revised version.

---

## Author Response (AR1)

Within this document, we respond to the comments of the reviewers. The structure is as follows:
Reviewer's comment in black
Our response to the reviewer's comment in blue
The added points/corrections applied to the manuscript in red.

**Reviewer #1**

The present study is concerned with the multidecadal satellite-derived Portuguese Burn Severity Atlas. The authors offer valuable insights into the topic. The authors present the following observations for consideration.

We thank the reviewer for their feedback and comments.

1. It is my contention that the role of small fires in the context of wildfire management is a significant yet understudied area. In contrast, the current study has a narrow focus on large fires (>100 ha).

We thank the reviewer for their insight. We focused on large fires (>=100 ha) because historically these correspond to the majority of area burned in Portugal. As stated by Fernandes et al 2009 and records from Divisão de Defesa da Floresta Contra Incêndios (DGRF), large fires correspond to only 1% of number of all fires but correspond to 75 % of the total burned area (mentioned in the manuscript in sub-section "2.1 Fire Data" lines 132 and 134). Concurrently, in our study, 68 % of all area burned is caused by fires >= 100 ha.

As highlighted by this comment, however, the exclusion of fires smaller than 100 ha is acknowledged as a limitation of our study. Our methodology can be applied to smaller fires (<100 ha) thus we encourage future studies to pursue execution of estimating burn severity for such fires. This point

is mentioned in the section "4 Discussion" in lines 446 to 448. The added point in the revised version is as follows:

Within this study, the focus was on large fires (>=100 ha), however, this can be considered as a limitation of our atlas. As our developed methodology can be applied to smaller fires (<100 ha), the exploration of burn severity of small fires can be a research opportunity for future studies.

2. It is anticipated that the fire atlas will have an identical resolution, resulting in comparable outcomes at either 30 or 500 meters. It is recommended that the ALI sensor be used for the 2012 data set.

We thank the reviewer for their suggestion. We agree with this comment and to accommodate it, in addition to MODIS-derived estimates for 2012, burn severity estimates via Landsat-7 ETM+ are included in the second version of the atlas which provide approximately 46 % of area of valid fires of 2012 with estimates. The rest (54 %) do not have estimates either due to gaps caused by the SCL failure or lack of cloud-free imagery. Users can now choose between having estimates for the year 2012 with:

-Landsat-7 data, with high resolution (30 m) covering about 37% of total burned area extent, or

- MODIS data, with moderate resolution (500 m), covering about 81% of the total burned area extent.

Aside from the Landsat series, images from Earth Observation-1 Advanced Land Imager (hereafter as EO-1 ALI) would have been the best option considering the spatial resolution. However, we have not created the atlas by manually gathering and processing the images, instead we have used "Google Earth Engine" and created a semi-automated process to gather, select, process, and calculate the burn severity estimates. Unfortunately, the EO-1 ALI dataset is not available on Google Earth Engine. Moreover, although EO-1 ALI sensor had the capability of having global coverage, it was an experimental satellite and mission-based, hence, it did not have continuous and frequent imagining with global coverage like the Landsat series of sensors and there are no images available for 2012 covering Portugal. As 112 large fires occurred in 2012 with the approximate extent of 81000 ha, there are not any available EO-1 ALI images. Nonetheless, manually gathering and processing images would be time consuming and is beyond the scope of the work.

This point has been added in the section "4 Discussion" in lines 491 to 504 as follows:

Burn severity estimates were obtained from different Landsat sensors to ensure "spectral consistency" over the long study period (Fernández-Guisuraga and Fernandes, 2024). This objective was achieved for all years. For 2012, aside from Landsat-7, estimates were also provided by MODIS Terra imagery. According to Landsat sensors' availability dates (Table1), no images from this family of sensors were available for this year aside from Landsat-7 ETM+ with SLC failure. One possible alternative would be to use images from Earth Observation-1 Advanced Land Imager (hereafter EO-1 ALI) available from November 2000 to March 2017 with a spatial resolution of 30 m (Chander et al., 2009). Although EO-1 had the capability of imagining with global coverage, it was an experimental and mission-based satellite (Hoang and Koike, 2018), and according to USGS website (EarthExplorer, 2025) there were no images available for Portugal in 2012. Hence, in addition to MODIS-derived estimates for 2012, burn severity estimates via Landsat-7 ETM+ are included in the second version of the atlas which provide approximately 46 % of area of valid fires of 2012 with estimates. The rest (54 %) do not have estimates either due to gaps caused by the SCL failure or lack of cloud-free imagery. Users can now choose between having estimates for the year 2012 with:
-Landsat-7 data, with high resolution (30 m) covering about 37% of total burned area extent, or
- MODIS data, with moderate resolution (500 m), covering about 81% of the total burned area extent.

3. It is my contention that a burn severity mapping system based solely on these indices with a fixed threshold will not yield optimal results. The resulting burn severity map is an inaccurate representation of the landscape due to the influence of environmental conditions, the diversity of objects, and the impact of climate.

We thank the reviewer for this comment. All our maps are presented in their continuous forms and no fixed threshold and classification have been applied on them. We have specified this point in the section "4 Discussion" lines 591 and 592.

We only presented the thresholds assigned by EFFIS to facilitate interpreting the maps by users if needed. However, the interpretation must be aligned by the users' objectives and context of the analysis. We have emphasized this point in the section "4 Discussion" in lines 578 to 579. Moreover, we have specified in the section "4 Discussion" lines 569

to 573, that any means of interpretation must be accompanied by field observations and users are required to take "caution" using any predefined means of interpretation. The reason for this caution is also added as follows in in the section "4 Discussion" lines 573 to 575 that reads:

This is crucial because burn severity often varies across vertical strata, and satellite-derived reflections are differently sensitive to impacts at each layer. Aggregating these impacts into a single metric can obscure important ecological details (Fernández-Guisuraga et al., 2023a; Miller and Thode, 2006; Parks et al., 2014; Cansler and McKenzie, 2012).

It must also be highlighted that the fixed thresholds by EFFIS are provided based on the comparison that they made between dNBR pixel values and ground burn severity assessment in the Mediterranean regions, considering the dominant environmental and climate conditions within these regions. This point will be added in the section "4 Discussion" in lines 580 to 583, as follows:

As an example of means towards interpretation of burn severity, in this study, the thresholds assigned by EFFIS are mentioned. The thresholds of EFFIS are assigned only for dNBR index and not for other indices and they are obtained from the comparison between dNBR pixel values and ground burn severity estimates considering the dominant environmental and climatic conditions within the Mediterranean regions (Llorens et al., 2021).

4. It is anticipated that the reliability of the atlas dataset will be enhanced by the incorporation of a more robust validation dataset.

We thank the reviewer for their comment. We have not used any "validation dataset" as we emphasized that no ground burn severity assessment was performed and no data in this regard were included nor analyzed (mentioned in the section "4 Discussion" in line 577 and 578). All these maps are estimates of burn severity and to assess their validity, they must be accompanied by ground burn severity assessment. The scope of this work does not include any validation. There are multiple studies in which the validation of different burn severity indices used in this atlas have been performed and they are mainly focused on different parts of the USA as mentioned in the section "4 Discussion" lines 567 to 569.

Examples of other studies with their focus on the Mediterranean regions will be added in the section "4 Discussion" lines 583 to 585. However, these studies are isolated and limited and hence, cannot be used to large scale evaluation. For example, the study of Fernández-García et al (2022) considered a total number of 23 fires among which only 4 fires were located in Portugal. We acknowledge the lack of validation as a limitation of this study, and we encourage future studies to perform more research in this regard. These added points will be in the section "4 Discussion" lines 585 to 594, that now read:

In Spain and specifically in Valencia province, by comparing CBI and dNBR, RdNBR, and RBR, the classification thresholds for interpretation of these indices have been introduced (Botella-Martínez and Fernández-Manso, 2017), which were furthered utilized for interpretation of burn severity estimates by ICNF for fires which burned in

Monchique and Portimão in Portugal (Anon, 2021). Moreover, there is a study conducted by Fernández-García et al (2022) considering the total number of 23 fires among which only four fires were located in Portugal (Fernández-García et al., 2022). Although there are studies comparing burn severity observations with estimates, they are isolated and limited and hence, they cannot be incorporated on large scale, which in our case is the mainland of Portugal. Hence, our maps have all been presented in their continuous raw forms and no classifications have been applied to them. We acknowledge the lack of validation as a limitation of our atlas, and we encourage conduction of future studies to validate the burn severity estimates of Portuguese Burn Severity Atlas.

5. It would be beneficial to examine the influence of seasonality and the climate dataset on the severity map.

We thank the reviewer for their suggestion. The analysis of burn severity drivers including climate variables was not within the scope of this study, while strongly encouraged by us to be performed in future studies. Our main goal is to provide users with burn severity estimates as a necessary foundation enabling future studies to perform all sorts of analysis specifically on the correlation of burn severity and the drivers (both top-down and bottom-up).

"Seasonality" as a burn severity driver was not explored as it was not within the scope of this study. However, the influence of "seasonality" on the quality of burn severity estimates, has been briefly explored as we have pointed out its role for instance by causing "mismatched phenology" between pre- and post-fire images (mentioned in the section "4 Discussion" within lines 530 to 541), while it was not quantified. Instead, we focused on the variation of the time lag and modelled it. Moreover, we tried to minimize the influence of the seasonality by capping the image acquisition sampling period, i.e., the duration to acquire satellite images, to ±120 days. By the analysis of the time lag variation from the first to fourth iterations, it was observed that the time lag on average does not exceed ±50 days (the corresponding result is presented in Figure B1 in the Appendix B section). However, the difference in seasonality may still have occurred and to minimize this impact, we have also calculated the burn severity offset values for all the indices and if the users are concerned with the seasonality variation, they can apply the offset values. Via the calculated offset values and comparing them to the time lag, the influence of seasonality can be also quantified. As the conduction of such analysis is encouraged by us, this point is added to "Discussion" section in lines 539 to 541. The added point is as follows:

Moreover, as suggestion for future studies, the lack of knowledge in the degree of seasonality influence on quality of burn severity estimates can be highlighted that via the offset values provided in the atlas and comparison between them and time lag such analysis can be conducted.

6. It would be advantageous to include the severity of different objects and trees during the fire period.

We thank the reviewer for their suggestion. Within this atlas, we applied images from the Landsat series of sensors with a resolution of 30 m. Hence, each pixel value corresponds to average reflectance of an area equivalent to 0.09

ha. Hence, distinguishment of burn severity of different land cover species and degree of burn severity of trees such as "crown damage" or "complete crown consumption" or in other words burn severity heterogeneity of different land cover types is not possible as this is a consistent limitation in capabilities of current sensors.

**Minor comments:**

1. It is recommended that all figures be enhanced in terms of quality.

We thank the reviewer for their comment. We will update the manuscript with higher quality figures. Moreover, in the revised version of the manuscript, the legend of Figure 5 was changed to "1st iteration (representing the blue boxplots).

2. It would be helpful to include a map indicating the frequency of burn for different areas.

We thank the reviewer for their suggestions. A figure is added to the Appendix A as Figure A1 to show frequency of valid fires within the parishes and districts of Portugal. Reference to this figure was added in the sub-section "2.1 Fire data" in lines 174 to 177. The added point is as follows:

To have an overview on the frequency of *valid* fires within the mainland of Portugal, FigA1 is provided in the Appendix. The frequencies of fires in Northern and Central regions are higher in comparison to other regions based on level 2 "Nomenclature of Territorial Units for Statistics" (presented as NUTS2) frontiers classification (Registo Nacional de Dados Geográficos - Direção-Geral do Território (DGT), 2024; Meneses et al., 2018).

[Figure]

**Figure A1** – Fire frequency of *valid* fires is presented within "Nomenclature of Territorial Units for Statistics" (NUTS2) frontiers (Registo Nacional de Dados Geográficos - Direção-Geral do Território (DGT), 2024; Meneses et al., 2018) demonstrating the extent of the mainland Portugal and its five regions.

3. It would be beneficial to analyze the impact of the threshold value for different sensors on the same fire.

We thank the reviewer for their suggestion. As Landsat sensors' availability dates do not overlap (Table 1), it is not possible to have burn severity estimates for one fire via images acquired from different sensors. Hence, although this analysis would have been indeed beneficial, its performance is not possible.

**Reviewer #2**

The data article provides a new dataset to potentially evaluate fire severity in Portugal in the period 1984-2022. This new dataset is a contribution to the analysis of fire severity in Portugal in the last decades. However, there are a series of limitations in the dataset that must be addressed prior to its publication:

The authors select fires larger than 500 ha for the period 1984-2000 to assess fire severity and fires larger than 100 ha for the remaining period, until 2022. The authors must justify the selection of these fire sizes and express in the methodology the percentage that these fires represent for Portugal in terms of number of fires as compared to the total number of fires in the country, and in terms of burnt area, as compared to the total burnt area in the country for all the years of the analysis. Fires larger than 100 ha are probably below 1% of the total number of fires, although they may have a larger contribution in terms of burnt area, maybe about 30-40%? These basic statistics are needed to know how representative the fire severity dataset is for Portugal. On the basis of the above and the availability of satellite imagery from Landsat sensors, the authors must justify the selection of the 500 ha and 100 ha thresholds. In principle and given that the imagery is at 30 m spatial resolution, there is no obvious justification for the selection of these thresholds.

We thank the reviewer for their comment.

The reason that we have assigned the minimum fire size to => 100 ha, as responded similarly to the first comment from the first reviewer, is due to the fact that historically and according to DGRF, these fires correspond to only 1% of number of fires while accounting for 75 % of burned area in Portugal (Divisão de Defesa da Floresta Contra Incêndios (DGRF), 2006; Fernandes, 2009). This point is mentioned in Section 2.1. Fire Data within lines 132 to 135.

Regarding the request to specify the statistics regarding fires>=100 ha, our own analysis revealed that according to fire data provided by ICNF, from 1984 to 2022, total area of 4.85 million ha is recorded burned with the total number of 37,581 fires. Within our study, we only consider *valid* fires which correspond to fires => 100 ha with known dates. The total sum of area of our valid fires is 3.92 million ha which corresponds to 68 % of all the fires recorded burned with total number of 5,099 fires accounting for 14 % total number of fires. These points are already mentioned in the manuscript within Section 2.1 Fire data in lines 170 to 174.

Burn severity estimates require pre- and post-fire dates and thus, estimates are sensitive to the time lag between fire and image acquisition dates. For the period 1984 to 2000, in our previous version of atlas, we were only able to find dates for fires => 500 ha for this period. Hence, we had no other option aside from assigning the minimum fire size to 500 ha for this period. However, after the submission of the manuscript we found that there were errors in the fire dates between 1984 to 2000 which resulted in errors in burn severity estimates for this period. Via analyzing satellite images, applying correction manually, and utilizing the combination of Monthly Fire Atlas (Neves et al., 2023) and the Forest Service fire datasets, the fire dates were corrected. This procedure allowed assigning fire dates to fires equal or larger than 100 ha, consistent with the rest of the years (2001 to 2022). We recalculated burn severity estimates for this period and integrated them in the second version of the Portuguese Burn Severity atlas. The manuscript is now updated with this new methodology as mentioned in Section 2.1 Fire data in lines 135 to 142. The updated points are as follows:

The fire perimeters were supplied by the Instituto da Conservação da Natureza e das Florestas (ICNF), (2021b). For the period from 1984 to 2000, uncertainties regarding the fire dates are greater than the subsequent years up to present. Hence, "Monthly Fire Atlas" (Neves et al., 2023) was used to provide dates for fires of this duration, using "day-ofyear (DOY)" dataset, which correspond to a band representing the day of year closest to the actual fire date of each individual fire. However, there were still cases in which multiple fires were marked as one resulting in inaccuracies regarding the dates. Hence, additional functions to analyze satellite imagery and manual corrections were implemented to discard any fires or proportion of their perimeters which did not have visible fire scar on false color composite (R: SWIR, B: NIR, and G: RED) image acquired on the date mentioned on "DOY" band or the fire scar also appeared on the image acquired prior to this date.

> Please, explain also why MODIS was used for 2012, which was a critical year of wildfires in Portugal and the potential inter-calibration (comparability) of the fire severity series for the rest of the years with the data obtained for 2012.

We thank the reviewer for their comment. We agree that 2012 is a critical year for Portugal from fire occurrence perspective. However, according to the availability of Landsat sensors (mentioned in Table 1), only Landsat -7 ETM+ imagery is available for this year. This sensor suffered from a technical issue in May 2003 in its scan line corrector (SLC) which resulted in stripes of gaps within its imagery. Due to these gaps and lack of cloud-free imagery, we were able to only calculate burn severity estimates for approximately 46 % of the burned area of valid fires. Thus, to allow having a larger burn severity extent, we provided MODIS-derived burn severity estimates with coarser resolution (500 m). In the second version of the atlas, the estimates with Landsat-7 imagery were included for the users to choose depending on their objective.

This point is added to Section 2.2 RS imagery: access and processing in lines 187 to 193 and it is as follows:

For 2012, there is no Landsat imagery available except for Landsat-7. This sensor suffered a technical failure in its scan line corrector (SLC) in May 2003 resulting in multiple gaps within its imageries since this time (Key and Benson, 2006). These gaps reduce the quality and availability of satellite imagery (providing 46 % of area of valid fires of 2012 with burn severity estimates). Hence, for only 2012, in addition to burn severity estimates obtained from Landsat-7, estimates from atmospherically corrected surface reflectance imagery of Terra abroad MODIS with spatial resolution of 500 m were provided. This addition aims to give users the options to choose between spatial resolution superiority of estimates with Landsat-7 or more areas with burn severity estimates via MODIS.

Regarding the comparability of estimates from Landsat and MODIS, Alonso-González and Fernández-García, 2021 stated that burn severity estimates derived from these sensors are comparable. However, they used estimates derived from Landsat -8 OLI in their study. Hence, we added additional exploratory analysis regarding the comparability of estimates derived from Landsat-7 versus MODIS. For this purpose, we created a subset from fires of 2012 and 2002 with total sum of area of 107,000 ha over 170 fires and correlated dNBR derived from these sensors following the methodology by Alonso-González and Fernández-García, 2021. Our analysis revealed that the correlation between MODIS and Landsat-derived dNBR is low (R=0.37). The corresponding part in this regard has been added to Section 2.2 RS imagery: access and processing in lines 193 to 196 and it is as follows:

In this regard, additional exploratory analysis was conducted to evaluate comparability of Landsat-7 versus MODIS derived dNBR- following the approach by Alonso-González and Fernández-García, 2021. Our analysis showed no correlation with details provided in Appendix A and Fig A2. Hence, for the statistics provided in this study such as sum of area with burn severity estimates, Landsat-7 derived estimates of 2012 were utilized.

The added results in Appendix A and Fig A2 are as follows:

To perform the exploratory analysis on comparability of Landsat-7 and MODIS, dNBR estimates of valid fires of 2012 and 2002 were used with the sum of area of 107,000 ha corresponding to 170 individual fires. The dNBR estimates were resampled to 500 m via "averaging" approach and the correlation was conducted following Alonso-González and Fernández-García, 2021. Our analysis showed that estimates from these two sensors may not be interchangeable due to weak and insignificant correlation (Pearson's correlation coefficient - R = 0.37 and the significance of correlation -P= 0) with estimates of MODIS having a tendency towards underestimation of durn severity. Fig A2 represents the obtained results.

[Figure]

**Figure A2.** The relationship between dNBR estimates via Landsat-7 and MODIS sensors are demonstrated via Gaussian kernel densities (in panel a) and scatterplot (in panel b) over the sum of area of 107,000 ha and over 170 individual fires burned in 2002 and 2012. Pearson's correlation coefficient is represented as R in panel b. The red line in panel b represents the linear regression performed on pixel-by-pixel correlation obtained via Landsat-7 (dependent variable) versus MODIS (independent variable) ($R^2$=0.13 and p-value =0).

The statement on the use of the DNBR-EVI regarding the potential use of this index based on the analysis of 3 fires, only, should be excluded from the article as the representativity of this analysis is not acceptable by any scientific standards. The authors are encouraged to continue the testing of this index with a sufficiently large/representative dataset.

We thank the reviewers for their feedback. This point is removed in the revised version.

---

## Referee Report (RR1)

General comments

I consider extended assessment to be approximately one-year post-fire, to coincide with the peak of the following year's green-up. It seems like too broad a range to use the definition of 2-12 months post-fire, unless the fire occurred in spring and thus peak green up followed shortly thereafter. In that case, pre-fire imagery should be used from the previous year's growing season. Perhaps it would be sufficient to add more explanation of the difference between rapid, initial, and extended assessments, in addition to pure timelines, such as what each assessment attempts to capture and represent. This might fit well after line 57 in the introduction.

Related to the above, your definition of burn severity could be more clearly framed throughout the paper. Because burn severity can refer to fire effects from immediate to one or even multiple years post-fire, it's important to define what your goals are and what your end product is representing.

I disagree with the approach of averaging all dNBR values for an area which burned more than once. Assuming a long time had passed since any prior fire, it would make more sense to represent the severity of the first fire. On the other hand, if the area is adapted to frequent fire, the severity of most recent fire entry might be more relevant. However averaging yields a value that does not represent the effects of any fire event and is thus ecologically meaningless.

I would suggest giving more context on vegetation types and historical fire regimes in the introduction if possible. This is very helpful when it comes up in the discussion but would be nice to have a general sense of earlier on, especially for readers not super familiar with fire in Portugal.

Line by line

Line 27 "climatic"

Line 48 add fuels, ie "changes on soil, vegetation, and fuels"

Line 53 "ash removal"

Line 115-119 don't need quotation marks

Line 134 does "fire date" refer to the start or end date of the fire? Or perhaps both, to fill in the Sdate and Edate values. Are progression data generally available?

Line 231 subject missing from sentence "in this case, ___ corresponds to images overlapping..."

Line 265 confusing grammar, missing something in this final sentence of the paragraph?

Line 470 "imaging" rather than "imagining"

Line 548 double negative, remove "no" or "neither"

Line 597-8 I suggest clarifying here the timeframe within which this finding is true (+/- 120 days).

---

## Author Response (AR2)

Within this document, we respond to the comments of the reviewers. The structure is as follows:
Reviewer's comment in black
Our response to the reviewer's comment in blue
The added points/corrections applied to the manuscript in red.

**Reviewer #3**

General comments

I consider extended assessment to be approximately one-year post-fire, to coincide with the peak of the following year's green-up. It seems like too broad a range to use the definition of 2-12 months post-fire, unless the fire occurred in spring and thus peak green up followed shortly thereafter. In that case, pre-fire imagery should be used from the previous year's growing season. Perhaps it would be sufficient to add more explanation of the difference between rapid, initial, and extended assessments, in addition to pure timelines, such as what each assessment attempts to capture and represent. This might fit well after line 57 in the introduction.

We thank the reviewer for this comment. The following has been added to the manuscript following line 57:

Beyond timeline differences, each assessment captures distinct stages of burn severity. For instance, delayed mortality and survivorship of vegetation are generally undetected during rapid assessment. Vegetation may still be senescent, stressed, or dying at this stage leading to under- and/or overestimation of burn severity. Exclusively, burn severity estimation during rapid assessment is normally performed to assist post-fire responses to large fires. However, during the rapid assessment, the influence of environmental responses to burn severity is minimized. Estimates obtained during the initial assessment are considered as the first opportunity to have a complete ecological evaluation as for instance, some signs of delayed mortality and survivorship of vegetation can be slightly detected during initial assessment while vegetation may still be senescent. During the extended assessment, environmental responses are the most influential on burn severity estimates while vegetation survivorship and delayed mortality are more easily detectable, and vegetation has normally returned to their stress-free green state. Overall, burn severity estimates obtained during the extended assessment are normally considered suitable as the final reference (Key, 2006).

Related to the above, your definition of burn severity could be more clearly framed throughout the paper. Because burn severity can refer to fire effects from immediate to one or even multiple years post-fire, it's important to define what your goals are and what your end product is representing.

We thank the reviewer for this comment. Our main objective was to provide estimates of the immediate fire impacts as our end product. However, due to the limitations of RS satellite imagery availability, we had to increase our sampling period to ±110 days. This point is now added throughout the manuscript as follows:
In Introduction section in line 130 the following has been added:

Thus, the main objective of this study is to create a high resolution multidecadal burn severity atlas for mainland Portugal entitled "Portuguese Burn Severity Atlas" — aimed at providing estimates of immediate fire impacts.

In 2.4 sub section entitled "RS imagery sampling period" line 259:

To address these issues, along with the objective of estimating the immediate fire impacts, we set our test sampling periods as follows: one day to 120 days…

> I disagree with the approach of averaging all dNBR values for an area which burned more than once. Assuming a long time had passed since any prior fire, it would make more sense to represent the severity of the first fire. On the other hand, if the area is adapted to frequent fire, the severity of most recent fire entry might be more relevant. However averaging yields a value that does not represent the effects of any fire event and is thus ecologically meaningless.

We thank the reviewer for their comment. We agree with your observation regarding the limitations of averaging dNBR values in areas that burned multiple times. For this reason, we have provided all annual severity maps in their absolute (non-averaged) form within the atlas. The averaged burn severity map shown in Figure 2, panel a, was included solely for illustrative purposes — to provide a high-level overview of general burn severity patterns across Portugal over the nearly 40-year study period. This map does not appear in our final atlas and is not intended to represent the burn severity of any specific fire event. Instead, it serves to offer visual context within the manuscript, supporting the broader temporal scope of our analysis.

> I would suggest giving more context on vegetation types and historical fire regimes in the introduction if possible. This is very helpful when it comes up in the discussion but would be nice to have a general sense of earlier on, especially for readers not super familiar with fire in Portugal.

We thank the reviewer for their comment. The following is inserted in the introduction in lines 121-125.

To the best of our knowledge, detailed long-term estimates of burn severity are missing for European countries, such as Portugal as the most "fire-prone" country in the European Mediterranean basin (Oliveira et al., 2011). Portugal is characterized by Mediterranean-type climate (Ermitão et al., 2023; Parente et al., 2023), a "drought-driven" fire regime (Pausas and Fernández-Muñoz, 2012), and dominated by shrubs and pines, eucalypts, and evergreen oaks forests (Fernandes et al., 2016). During the past decades, Portugal has been significantly affected by fires, such as catastrophic fires in 2003, 2005, and 2017 (Nitzsche et al., 2023; Beighley and Hyde, 2018). Thus, the main objective of this study is to create a high resolution multidecadal burn severity atlas for mainland Portugal entitled "Portuguese Burn Severity Atlas" — aimed at providing estimates of immediate fire impacts.

Line by line

We thank the reviewer for their specific corrections. All of the following corrections are applied accordingly.

> Line 27 "climatic"

Climatical has been converted into climatic.

> Line 48 add fuels, ie "changes on soil, vegetation, and fuels"

It is added.

Line 53 "ash removal"

"ashes wash off" is now changed into "ash removal"

Line 115-119 don't need quotation marks

The quotation marks are removed.

Line 134 does "fire date" refer to the start or end date of the fire? Or perhaps both, to fill in the Sdate and Edate values. Are progression data generally available?

The progression data are not provided by the forest service of Portugal (ICNF) and we only have access to date of alert and containment date of fires, which we converted to Sdate and Edate. After our analysis, we understood that the recorded Sdates and Edates were not accurate specifically for fires prior to 2000 and we applied additional measures to correct them. In this line, fire date corresponds only to Edate. We updated this line to the following:

… which correspond to a band representing the day of year closest to the Edate of each individual fire.

Line 231 subject missing from sentence "in this case, ___ corresponds to images overlapping…"

Here, the missing subject was "IC" which is now added. Hence the sentence is updated as follows:

In this case, IC corresponds to images overlapping…

Line 265 confusing grammar, missing something in this final sentence of the paragraph?

The sentence is changed to the following:

From the found correlation and adaptation of our sampling period, a function represented in Eq. (1) was developed to calculate "SUITABILITY" property, which penalizes potential RS images based on their time lag —assigning values from 100 (for an image with a 0-day lag) to 0 (for an image with a 111-day lag).

Line 470 "imaging" rather than "imagining"

We thank the reviewer for this comment. We converted "imagining" to "imaging".

Line 548 double negative, remove "no" or "neither"

We removed "neither'.

Line 597-8 I suggest clarifying here the timeframe within which this finding is true (+/- 120 days).

The sampling period applied to create Portuguese Burn Severity Atlas is now mentioned in this sentence as follows:

With 97 % of burn severity estimates of valid fires from 1984 to 2022 within the sampling period of ±110 days, we can … .